# Zeolites as Ingredients of Medicinal Products

**DOI:** 10.3390/pharmaceutics15051352

**Published:** 2023-04-28

**Authors:** Iane M. S. Souza, Fátima García-Villén, César Viseras, Sibele B. C. Perger

**Affiliations:** 1Laboratório de Peneiras Moleculares, Universidade Federal do Rio Grande do Norte, Natal 59078-970, Brazil; 2NanoBioCel Group, Faculty of Pharmacy, University of the Basque Country UPV/EHU, 01006 Vitoria-Gasteiz, Spain; 3Department of Pharmacy and Pharmaceutical Technology, Faculty of Pharmacy, University of Granada, Campus Cartuja s/n, 18071 Granada, Spain; 4Andalusian Institute of Earth Sciences, CSIC-University of Granada, Armilla, 18100 Granada, Spain

**Keywords:** zeolite, drug delivery system, modified drug release, novel formulations

## Abstract

Development of new medicinal products for particular therapeutic treatment or for better manipulations with better quality and less side effects are possible as a result of advanced inorganic and organic materials application, among which zeolites, due to their properties and versatility, have been gaining attention. This paper is an overview of the development in the use of zeolite materials and their composites and modifications as medicinal products for several purposes such as active agents, carriers, for topical treatments, oral formulations, anticancer, the composition of theragnostic systems, vaccines, parenteral dosage forms, tissue engineering, etc. The objective of this review is to explore the main properties of zeolites and associate them with their drug interaction, mainly addressing the advances and studies related to the use of zeolites for different types of treatments due to their zeolite characteristics such as molecule storage capacity, physical and chemical stability, cation exchange capacity, and possibility of functionalization. The use of computational tools to predict the drug—zeolite interaction is also explored. As conclusion was possible to realize the possibilities and versatility of zeolite applications as being able to act in several aspects of medicinal products.

## 1. Introduction

A drug or “active substance” is a bioactive molecule that, in view of its structure and/or chemical properties, is able to produce one or more biological effects (called therapeutic effects) by means of its interaction with certain macromolecules (receptors) which are in the “active site”. Even if a drug could be administered as a pure molecule, this rarely occurs for a wide range of reasons. Therefore, medicinal products are not pure chemicals but a combination of different ingredients (the so-called “excipients” together with one or more active substances). While, as mentioned, drugs are intended to produce a particular therapeutic effect, the excipients are inactive molecules with a wide range of functions within the medicinal product. That is, excipients are additives associated with active substances to act as carriers, facilitate their preparation and manipulation, improve stability, modify the organoleptic or physicochemical properties, increase bioavailability, etc. Moreover, excipients allow the drug to be administered through different routes depending on the drug properties itself but also on the site of action or the desired therapeutic effect. A dosage form is, in fact, the arrangement to which the active ingredients and the excipients are adapted to constitute a medicinal product. More particularly, a drug delivery system (DDS) can be defined as a dosage form that enables a drug to selectively reach its site of action and minimizes the drug contact with non-targeted cells, organs, or tissues.

Until the 1950s, a major portion of drugs was administered as pills or capsules with no ability to control drug release. The administration of drugs in their conventional dosage forms can in some cases lead to impaired drug bioavailability and plasma drug level fluctuations, which usually results in unwanted toxicity and compromised efficacy. In 1952, the first prolonged release technology, called Spansule^®^, was produced [1]. Spansule^®^ formulation consists of the formation of pellets composed by the drug coated with various thicknesses of a selected coating with different slow dissolution and placed into capsules. In this advanced DDS, the coating excipients control the drug release since they limit the access of fluids to the drug molecules inside the sphere/bead, something that varies depending on the dissolution rate and thickness of the coating itself [2,3]. From that moment on, the number of studies and formulations able to control and improve drug performances have significantly increased, and the concept Modified Drug Delivery System (MDDS) was also introduced. The scope of MDDSs is to carry the drug to the action site with a proper rate and concentration, to minimize or eliminate side effects, prolong the drug action, reduce posology, etc. In short, MDDSs attempt to improve drug bioavailability, thus maintaining effective drug concentration in the action site as long as possible, with minimal side effects [3,4,5]. Thus, MDDSs are basically intended to respond to particular and specific problems of certain drugs or dosage forms. Therefore, in the design of a MDDS, it is necessary to carefully arrange and select the ingredients. When it comes to the drug, its molecular weight, size, water solubility, lifetime, and administration route must be considered [3]. During the selection of the excipients, certain aspects must be taken into account that would be determined by the main scope of the MDDS. For instance, their particle/molecule size, shape, structure, functional groups, specific area, solubility, biocompatibility, toxicity, physicochemical properties, possibilities of functionalization, and surface modifications, not to mention that all the excipients must comply with the minimum standards established by the pharmacopoeias, something decisive for the subsequent commercialization of the medicinal product [3,6,7].

In this regard, natural inorganic aluminosilicates such as clay minerals have played a crucial role. They have been widely used in pharmaceutical and cosmetic formulations as both excipients and actives. For many years, clays such as talc, kaolin, and montmorillonite (often known as bentonite), among other things, have been in the spotlight of pharmacy due to their physicochemical and textural properties. In fact, there is a remarkable number of research papers and reviews dealing with clay mineral roles in the formulation of DDS and MDDS [7,8,9,10]. Zeolites, especially those of natural origin, have recently emerged as potential materials to be explored in pharmaceutical and biomedical applications [11,12].

Zeolites are also aluminosilicates, just like clays, though their three-dimensional structures are quite different. These differences are not the scope of this review since this document does not intend to deal with clays. The discussion merely focuses on zeolites, which are hydrated crystalline microporous aluminosilicates. A zeolite’s structure is mainly composed of a three-dimensional network of [SiO_4_]^4−^ and [AlO_4_]^5−^ tetrahedra connected to each other through oxygen atoms. The different orientations of tetrahedra give rise to a wide range of crystalline structures with internal mesopores, micropores, and channels along the entire crystalline structure. This distinctive feature allows zeolites to act as molecular sieves, something that has been particularly useful for catalysis, separation, and purification of molecules [13].

The occurrence of silicon isomorphic substitutions by other elements such as aluminium creates a charge imbalance in the structure which is compensated by exchangeable alkali and alkaline earth metal cations, located either in the outer surface or within the pores/channels. Other positive molecules such as ammonium ion can also be associated with zeolites to compensate the aforementioned charge imbalance. Figure 1 shows the structure of a zeolite and some elements that may generally be present in its composition such as Al atoms, water molecules, and cations. In fact, the Si/Al ratio is one of the most determinant features of a zeolite since it defines its hydrophilic/hydrophobic nature, acid stability, and exchange capacity. These properties, together with others such as the high sorption capacity and porosity, their regular and uniform pores, and their thermal and chemical stability make zeolites suitable as drug carriers and, therefore, particularly useful in the design of MDDS. What is more, zeolites can be synthesized in the laboratory under controlled conditions, thus allowing the obtaining of zeolites with particular characteristics depending on their usage, not to mention that the resulting product possesses a remarkable purity with respect to the natural counterpart.

As it happens with other ingredients such as clay minerals, the physicochemical properties of zeolites can be controlled and modified to be used in pharmacy and biomedicine [11,14,15,16]. In particular, zeolites have been gaining a lot of attention in the pharmaceutical field due to their ability to establish effective interactions with a wide variety of drugs. Moreover, the functionalization of the surface of zeolites could neither be overlooked since it widens the range of possible MDDS.

### Factors Determining the Role of Zeolites as Drug Carriers

The adsorption of drugs onto zeolites is a complex process in which different factors and conditions must be considered to maximize the process. As an example, the adsorption of ketoprofen, hydrochlorothiazide, and atenolol, three very different drugs, were evaluated in different types of zeolites [17]. This study revealed that conditions such as pH, Si/Al ratio, exchangeable elements, and thermal treatment of zeolites significantly influence the interaction between both species. It was concluded that the adsorption capacity of the beta zeolite greatly depends on the pH and the amount of Al in the zeolite. Moreover, zeolite samples that were previously calcined before the adsorption process showed a significant increase in the final amount of drug adsorbed. This result can be ascribed to the complete removal of adsorbed water, organic species such as ammonia, and other residues from the zeolite.

Thanks to the extensive literature already published in this field, it is possible to summarize the main factors and zeolite properties to bear in mind when preparing a zeolite-based MDDS, as shown schematically in Figure 2.
−Si/Al ratio. Low Si/Al ratios have demonstrated to favour the adsorption of drugs via hydrogen bonds. This have proved to be important with drugs such as acetyl salicylic acid (aspirin) [15]. Extreme values of either Si or Al (Si/Al ratios close to 1) compromise zeolite stability in certain conditions such as acid environment, not being appropriate for oral administration. The Si/Al ratio of zeolites can be modified by using different techniques such as acid treatments [18,19]. This modification causes a dealumination of the zeolite framework, thus decreasing the number of charged sites in the mineral [20]. An example about the influence of Al on zeolite as a carrier was the study with probucol (a hydrophobic oral drug), which performed a higher release rate from zeolites with low aluminium content (zeolite beta) with respect to other zeolites (zeolite NaX) with higher amounts of aluminium [21].−The hydrophilicity of the zeolite, intimately related to the Si/Al ratio, has demonstrated not only to determine the adsorption capacity but also the release mechanism. High Si/Al ratios imply lower hydrophilicity. A high hydrophilicity has been associated with a greater release of drugs in vitro due to a faster hydration and contact with water molecules, which favours the dissolution and release of the corresponding active [15,17].−Isomorphic substitutions. The substitution of Si for elements other than Al such as Be, B, Ga, Ge, or P has also demonstrated to influence drug–zeolite interactions. Zeolite beta was synthetized with different boron substitutions to study the resultant structure and porosity as well as the ability to adsorb thiamine [22]. Neither porosity, texture, nor drug-sorption properties changed when all the aluminium was replaced by boron, whereas the simultaneous presence of both elements (Al and B) in the zeolite beta framework gave rise to higher specific surface area and pore diameters. The highest thiamine adsorption capacity was performed by the zeolite containing boron. Authors reported that the zeolite containing boron showed lower Brønsted acidic sites and higher amount of Lewis acid sites with respect to the Al-zeolite, which can lead to greater interactions between the zeolite surface and the drug for this case.−Pore size and geometry. According to the International Union of Pure and Applied Chemistry (IUPAC), pores are subdivided according to their internal diameter into micropores (>2 nm), mesopores (between 2 and 50 nm), and macropores (<50 nm) [23], in which zeolites generally present pores between the range of micro to mesopores. Depending on the drug molecular size, this pore opening and the geometry of the channels must be sufficient to allow access and diffusion within the zeolite structure in order to favour the encapsulation and release of different actives. In addition to porosity, surface area is an important parameter when it comes to adsorption and release of drugs. According to Esquivel–Castro et al., the increase in surface area is proportional to the amount of the drug stored in core/shell nanoparticles [24]. These properties make it possible to control the release and/or to protect certain molecules such as nucleic acids, corticosteroids, or anticancer drugs [25,26,27,28]. Nonetheless, these properties on their own usually fail to predict the drug–zeolite interaction; thus, it is necessary to consider the flexibility of the drug molecules and the zeolite framework in different conditions (such as high temperatures). In particular, Wernert and co-authors [29] observed that during hydrothermal treatment the pore volume of faujasite decreased in a directly proportional manner to the temperature, which subsequently influenced the entrance of the drug in the micropore. The use of structure-directing agents is another versatile mechanism for zeolite synthesis that uses a template molecule, organic or inorganic, as a model during the crystallization process, enabling the development of adequate porosity for the synthesized zeolite, creating a regular chain of pores and channels. After the crystallization, the template is removed. For instance, the addition of p-cresol during zeolite faujasite-Y synthesis resulted in a zeolite having an optimal pore size to retain this particular molecule (and smaller ones) [30]. In addition, several methodologies can be applied before or after the synthesis of the zeolite in order to create a secondary porosity, obtaining as a result hierarchical pores, expanding the versatility of the use of zeolites even more [31].−Particle size. The smaller the zeolite particle size, the higher the specific surface, and, therefore, the higher the surface available for possible drug–zeolite interactions [27,28,32]. The particle size is of great interest when the active sites interacting with the active molecule are on the surface of the zeolite rather than in the zeolitic channels and pores [29].−Cation exchange capacity (CEC) and cations affinity. The CEC allow zeolites to establish electrostatic interactions with cationic molecules by exchanging their naturally associated cations (such as Ca^2+^, Na^+^, Mg^2+^…) and positively charged molecules (ammonia and nitrate ions) with new cations or other positively charged molecules located in their surroundings [33]. Ion exchange is one of the most important property of zeolites. The isomorphic replacement of Si by elements with different charge, Al for example, become zeolite charged, which is counterbalanced by ions, called counterions. When the zeolite is immersed in a phase containing others ions in higher concentration the diffusion process is started, the initial ions diffuse out from the framework and the news ions diffuse into the solid structure [34]. This property causes zeolites to be used for the removal of toxins from wastewater, and they can be also explored for medicinal application. In fact, this property has been associated with antacid activity of clinoptilolite [35], which is able to exchange cations with the H^+^ of the gastric fluids. The affinity by certain types of cations should also be considered since it will determine the mobility of chemical species [36]. Moreover, different cations could alter the adsorption/desorption capacity of organic molecules. As an example, two potassium cations would be replacing one calcium, which could explain the variability in the adsorption capacity of zeolites with different exchangeable cations [29].−Origin: natural or synthetic. The use of naturally occurring zeolites implies the presence of different minerals and impurities, as well as possible contaminants that must be eliminated (or at least reduced) before their use in the pharmaceutical field. In 2016, Cerri and co-workers published an article on the characterization and purification of clinoptilolite in order to accomplish the requisites of the Japanese and European Pharmacopoeias [14]. Moreover, the purity and richness are also crucial to maximizing the interaction with drugs [33]. The synthetic zeolites can be advantageous due to the possibility of obtaining zeolites with predetermined properties depending on the synthesis process and with less impurities. The synthesis process consists of preparing a mixture containing all the specific components correlated to each zeolite structure and submitting this initial precursor mixture to hydrothermal treatment inside an autoclave at a specific temperature to enable the nucleation and growth of zeolite crystal [37].−Zeolite surface modifications. Since not all the drug molecules are prone to interact with pure zeolites, surface modifications of the inorganic ingredient can be performed. In this regard, surfactants are commonly used in the modification of minerals. To better understand this point, for example, sulfamethoxazole is less polar than metronidazole, and both antibiotics were combined with clinoptilolite after cationic surfactant modification. The results showed that the adsorption of sulfamethoxazole was enhanced by the presence of the amphiphilic molecule. On the other hand, the uptake of metronidazole was independent of the surfactant presence [20]. Therefore, the use of surfactant can also affect the nature of the drug–zeolite interactions, something that can be very useful toward optimizing the drug release process [32]. These observations can also be extended to other surface modifications or excipients combinations such as zeolite/polymer nanocomposites.

**Figure 2 pharmaceutics-15-01352-f002:**
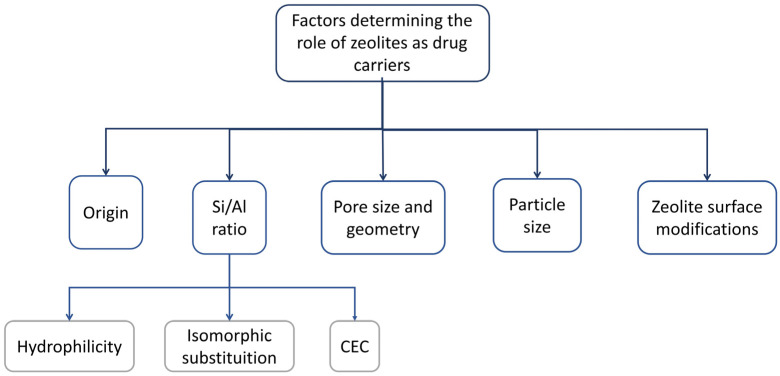
Overview and classification of the main factors determining the selection of zeolites as drug delivery systems. Original reproduction.

Cytotoxicity is one important parameter to be considered in the application of new materials as ingredients for medicinal products. According to Bacakova et al. [38], some zeolites present an undesirable cytotoxicity such as the case of erionite, but in some cases this cytotoxicity can be advantageous for the treatment of tumors, that is, as well as several other parameters, a possible cytotoxicity of a zeolite must be studied case by case [38]. Studies report low toxicity for many types of zeolites such as the adsorption, release, and cytotoxicity of zeolite Y nanoparticles for the cisplatin, an anticancer medicine, demonstrating that the nano-zeolite did not show toxic effects on MG63 cells (supplemented with fetal bovine serum and streptomycin-penicillin) and exhibiting a good biocompatibility. Additionally, the composite ZC-NPs made by drug encapsulation showed a high cytotoxicity and an important reduction in cell viability [39]. The cytotoxicity activity of nano-zeolite Y and A using alveolar epithelial cells (A549), human endothelial cells (EA.hy926), and differentiated macrophages (THP-1) cell lines by mitochondrial activity (MTT) and cell membrane integrity (LDH leakage assay) were reported and demonstrated no significant cytotoxic after 24 h of exposure [40]. Additionally, no significant toxicity was observed for zeolites LTL and EMT [41], ZSM-5, zeolite A, and Faujasite NaX [42].

## 2. Zeolite as Actives

In the same way that clay minerals can act as active ingredients on their own, zeolites, have been shown to possess certain therapeutic activities by themselves. The use of zeolites in veterinary medicine is one of the most widely known and studied uses of zeolites [43]. Nevertheless, this section is strictly dedicated to reviewing and discussing the use of zeolites as actives for human use according to their mineral’s origin (natural or synthetic), which is of great importance due to the differential purity and richness.

### 2.1. Natural Zeolites

Clinoptilolite is highlighted as one of the most abundant, naturally occurring zeolites. The tasajera deposit (Cuba) is one of the most famous exploitations of natural clinoptilolite worldwide. This zeolite has been exploited as gastrointestinal active ingredient since 1995, when the Cuban Drug Quality Control Agency approved its use as a human anti-diarrhoeic drug (Enterex^®^) [44]. Its anti-diarrhoeic mechanism of action is related to the high adsorption capacity and specific surface area of this aluminosilicate. These properties allow for the adsorption of excessive bile acids, glucose, bacteria, and protozoa toxins, these being some of the etiological agents of diarrhoea. From that moment on, purified clinoptilolite from Tasajera deposit has also been registered as antacid drug (Neutacid^®^) and administered as chewing tablets [35]. In this case, the CEC of clinoptilolite was responsible for the antacid therapeutic activity.

The antiviral activity of clinoptilolite was also evaluated by Grce and Pavelic [45]. They tested the antiviral properties of this zeolite against human adenovirus 5, herpes virus type 1, and enterovirus Coxsackie B5 virus and echovirus 7. The results indicated that high doses of clinoptilolite were needed for an effective treatment, thus hindering its pharmaceutical application. Nonetheless, authors reported the possibility of using clinoptilolite as a drinking water purification ingredient, thus reducing their transmission through drinking water. As far as we are concerned, no other antiviral uses of clinoptilolite have been reported.

Another useful therapeutic activity of zeolites, in general, is their ability to adsorb molecules with dimensions capable of accessing its pores and cavities. In particular, clinoptilolite has been part of Panaceo^®^ sport, a formulation intended to adsorb NH_4_^+^, CO_2,_ and H_2_S from protein-rich diets [46]. In this regard, one of the most recent studies deals with the transport of gases by zeolites to glioblastoma tumours [47]. Being that this is one of the most aggressive cancers to which no effective treatment has yet been found, all the possible approaches must be considered. In this regard, the study by Anfray and co-workers [47] focused on the oxygenation of the tumour in order to increase the oxygen pressure selectively into the tumour and the response to traditional treatments. Na-Faujasite and Gd-Faujasite were tested as an O_2_ and CO_2_ carriers in a rodent model.

Other toxicants such as mycotoxins, aflatoxins, zearelenone, ochratoxin [48], lead [49], and organophosphate [50] are effectively adsorbed by clinoptilolite when included in animal diets, thus reducing the morbidity associated to intoxication in livestock.

Since cancer is currently one of the diseases with higher impact in society, the attempts of the scientific community to treat it are numerous. In this regard, zeolites have also been evaluated as an anticancer drug on their own. The study of Pavelić et al. [51] revealed that clinoptilolite can induce the expression of tumor-suppressive proteins, thus compromising and preventing the cancer cells growth and proliferation. Three years later, Zarkovic et al. [52] studied the anticancer effects of micronized clinoptilolite. The experiments, performed in various tumorous cell cultures and tumour-bearing animals, revealed the usefulness of this mineral. Moreover, micronized clinoptilolite, in combination with doxorubicin, reduced pulmonary metastasis.

Natural zeolites have also proved to be effective during haemostasis. Nowadays, it is well known that, thanks to the elevated porosity of zeolites, they can act as oxygen reservoirs and platelets activators. Consequently, they can accelerate the clot formation and wound scar formation [38,53,54]. Natural clinoptilolite and mordenite from Jinyun, China, were compared to QuikClot^®^ (synthetic zeolite) in a rabbit model with complex joint injury [55]. The natural clinoptilolite and mordenite were promising since they not only provide clot formation, but also improved the wound-healing process with respect to QuikClot^®^. These results agreed with those previously obtained by [56], which revealed the wound-healing potential of natural clinoptilolite. The zeolite was previously activated by a centrifugal process on a tribomechanical micronization and activation device, which produced nanoparticles with a total surface of 50,000 m^2^ and a particle size of 200 nm. In this occasion, clinoptilolite acted as an immuno-stimulator that helped scar tissue formation. Additionally, UV and antibacterial protection was also provided by clinoptilolite, which are very convenient performances of an ingredient during the wound-healing process [56,57].

A high surface area clinoptilolite (Panaceo Micro Activation or PMA-zeolite) was studied on the treatment of osteoporosis, a major public health problem with several complications [58]. This PMA-clinoptilolite was administrated for 12 months to female rats ovariectomized to induce an osteoporotic condition. At the end of the study, the left femur and tibia were analysed to compare the effectiveness of the zeolite oral treatment. The PMA-clinoptilolite demonstrated to promote bone formation and reduced bone resorption after ovariectomy. In this study, all clinoptilolites reported to be useful to adsorb different elements in the intestine, as well as to reduce the Al blood levels of osteoporotic rats. These results encouraged the authors to conduct a clinical study with 100 osteoporotic patients ranging 56 to 74 years old [59]. All these patients had a bone mineral density of 2.5 or lower at the beginning of the 12-month treatment. The clinical results showed a significant improvement in the patients without significant adverse effects. Moreover, clinoptilolite has a low-binding affinity to anti-osteoporotic drugs such as bisphosphonates. This, together with the independent action of the PMA-clinoptilolite, led authors to state that PMA-clinoptilolite “presents as a potentially promising alternative or adjuvant therapy for osteoporosis [59]”.

### 2.2. Synthetic Zeolites

A major portion of currently existing zeolites are of synthetic origin [60]. Zeolite synthesis consists in general of a hydrothermal crystallization process in alkaline medium. The preparation of an initial precursor mixture containing the aluminosilicate sources enables the first polymerization and depolymerization processes, and the hydrothermal step leads the nucleation and crystallization process where the source under action of mineralizing agents and the structure-directing agent (SDAs) transforms into the zeolite crystal, as shown schematically in Figure 3 [37]. The optimization of different parameters such as temperature conditions, agitation, and Si/Al ratio, among other things, allows the synthesis of different zeolite structures [61,62].

In terms of pharmaceutical applications, synthetic zeolites are more advantageous since it is possible to control/minimize the presence of impurities (such as unreacted chemicals or associated crystalline structures) and to obtain a controlled morphology with the desired properties for a specific drug [15]. This section focuses on the roles of synthetic zeolites as therapeutic ingredients.

Synthetic zeolites are also useful as antacid drugs. Absorbatox^TM^ 2.4D is an “artificially enhanced” zeolite that has proven to possess gastroprotective properties against negative gastroesophageal reflux disease and nonsteroidal anti-inflammatory drug-induced gastritis [63]. Although Potgieter et al. [63] claimed that the exact mechanism of action is not clear enough, they hypothesized that it could be related to the adsorption of H^+^ in the stomach and its exchange by zeolite cations, just as it happens with clinoptilolite.

QuikClot^®^ is the most well-known zeolite-based haemostatic product. It was patented and commercialized as a combat gauze indicated against external haemorrhages of soldiers. It is composed of zeolite that promotes clot formation through water adsorption, which is favoured by its high specific surface area [64,65]. Nonetheless, the application of QuikClot^®^ produced an exothermic reaction that limited its use. Notwithstanding its disadvantages, QuikClot’s discovery opened a field of study to search for new natural inorganic, haemostatic ingredients [55,66]. Due to its adhesiveness, biodegradability and biocompatibility chitosan is another typical product for haemostasis in extreme conditions. HemCom^®^ is a US Food and Drug Administration (FDA)-approved chitosan dressing [67]. Based on this product and the usefulness of zeolites as haemostatic product, Zhang and co-workers prepared and explored the use of the biodegradable polymers gelatine, chitosan, and alginate combined with an unspecified zeolite to obtain an antibiotic-loaded composite with haemostatic and antibacterial activity [68]. The aim of this study was to either enhance the clot formation as well as to prevent common subsequent infections. The combination of the inorganic component (zeolite) with polymers was performed through the inversed emulsion method, which led to hollow microspheres with three pores sizes: nanopores from zeolite structure, pores between zeolite particles, and void cores from the microspheres. This exceptional porosity brings on a significant water absorption capacity, which is an important feature in triggering haemostasis. Later, a zeolite-cotton patch was proposed as haemostatic medical device by Yu and co-workers [69]. On this occasion, chabazite was synthesized in cotton fibres, thus creating a composite gauze that promoted clot formation in a short time without exothermic reaction. Recently, ZSM-5 and silicalite-1 zeolite were tested as against *C. auris*, a multidrug-resistant microorganism, and the results showed a considerable inhibitory effect [70].

## 3. Zeolites as Drug Delivery Systems

The usefulness of zeolites as platforms for drug transport and release is especially interesting due to the versatility of these minerals, which mainly derive from their intrinsic properties. Thus, zeolites have proved very useful as drug carriers (by simple physical adsorption), as molecular sieves (due to their porosity), as drug DDS and MDDS thanks to the different zeolite–drug interactions, and the difference in the location of the active substances (zeolite surface, within the zeolite pores and channels…). In addition, they can be combined with other substances to be grafted, modified, or functionalized. They are also useful for reinforcing the mechanical properties of other labile substances such as polymers, something of special importance not only in the formulation of MDDS but also for other fields such as tissue engineering. The possibility of synthesizing zeolites allows the controlling of many final parameters such as pore/channels’ size (useful to adapt them to the final scope of the formulation), final purity, and particle size, thus allowing parenteral administration of these inorganic ingredients.

### 3.1. Topical Treatments

Inorganic aluminosilicates such as clay minerals have been traditionally used for topical dosage forms and formulations either as cosmetic or as medicines. Due to their similar chemical composition and properties, zeolites are equally useful ingredients of topical formulations. Among the most explored topical uses of zeolites, the carry and controlled release of antimicrobial actives can be mentioned. For instance, natural clinoptilolite was evaluated as a carrier for Zn^2+^ and erythromycin with the objective to treat acne [71]. In view of the positive results and effectiveness, the composite was fully characterized and patented [72].

Synthetic ZSM-5 zeolite was synthetized through hydrothermal route and proposed as a gentamicin drug delivery system by Guo et al. [73]. Gentamicin is an antibiotic designed to treat skin infections, and it has a molecular size compatible with ZSM-5 micropores. In fact, the loading and placement of the antibiotic within ZSM-5 micropores allowed for a gentamicin control release [73].

Silver, sulfadiazine, and the combined drug substance “silver sulfadiazine” (AgSD) were the topical antibacterial molecules chosen to treat infections in which prophylaxis is determinant (such as burns). In fact, AgSD is commonly used in the treatment of burns, wherein a cream is applied onto the burned area while protected with bandages. Nevertheless, the cream could dry over time and the bandages require frequent replacement. To improve the treatment of burns and knowing the usefulness of zeolites as DDS and MDDS, researchers studied the development of a chitosan/AgSD/zeolite NaY composite film for wound dressing. The NaY zeolite was impregnated with AgSD, and then added to a chitosan solution for the film preparation [74]. The resultant composite film proved to be effective against *Candida albicans* fungus and different bacterial strains. Moreover, the zeolite promoted a sustained release of the silver ions for longer times. However, according to the cytotoxicity studies, the final formulation needed to be optimised [74]. One of the pure AgSD drawbacks is that it is practically insoluble in water and its antibacterial activity occurs during the dissociation of silver ions under physiological conditions. In a recent study, a beta zeolite was proposed as a drug carrier of both silver and sulfadiazine to provide both actives already dissociated, therefore facilitating the mechanism of action [75]. Zeolite-beta was loaded with silver by the solid-state ion exchange method, whereas high-energy ball milling was used to load sulfadiazine. Sulfadiazine release was prolonged by the presence of Ag in the zeolite. Moreover, silver ions were also released together with the drug, thus explaining the synergic bioactivity of the composite in comparison with pure silver sulfadiazine.

In the particular case of topical wound-healing treatments, a controlled drug release is crucial to guaranty a long-lasting activity and to reduce the posology, which subsequently avoids the necessity to excessively manipulate the wound, something that could hinder or interrupt the healing process. A polymeric hydrogel of polyvinyl alcohol and chitosan, reinforced with clinoptilolite, has been recently prepared [76]. The zeolite was added to study its influence in the wound-dressing mechanical and drug release properties. Moreover, the dressing was also loaded with γ–cyclodextrin/thymol composite. Thymol is a natural, essential oil extracted from *Thymus serpyllum* and *Thymus vulgaris*. It shows remarkable antibiotic activity, thus resulting in a potential natural candidate for the treatment and prophylaxis of infected wounds. The addition of clinoptilolite improved the mechanical properties of the hydrogels, including swelling capacity and the water–vapour transmission rate. Clinoptilolite wound dressings showed a more controlled thymol release, something associated with a more compact dressing structure [76].

The design of hydrogel films based on natural polysaccharide low-methoxyl (LM) pectin and zeolite A was studied to apply in topical DDS, evaluating, among other parameters, the effect of the zeolite cation (Na^+^ or Zn^+^) on the theophylline release [77]. In fact, the drug release was affected by the cation present in the zeolite: films prepared with Zn-zeolite composite had lower swelling ratio than Na-zeolite composite, something ascribed to the valence of the cation. In addition, the Zn-zeolite composite was able to release Zn ions that could act as an adjuvant in wound healing and provide antimicrobial effects. On the other hand, the Na-zeolite composite would be more prone to promote cell viability and proliferation during the healing process. Afterwards, the authors reported the synthesis of pectin-based hydrogel films with the ability to deliver albumin in a controlled manner. The hypoalbuminemia is a disorder caused due to the low albumin production, which can be related to chronic wound complication, wherein the albumin infusion and protein administration are some of the treatments practiced in the wound of hypoalbulminemia. Similarly, as in the previous publication, the authors evaluated zeolite-A on the composition of the hydrogel film to improve the oxygen permeability and to stall the decomposition of the pectin hydrogel. The adsorption results showed a pH-sensitive behaviour in the adsorption of bovine serum albumin (BSA) on the zeolite and a successfully adsorption in pectin zeolite matrices at an adequate pH. The presence of zeolite on the pectin hydrogel increases the BSA adsorption; nevertheless, no release was observed, which the authors related to an insufficient driving force for the diffusion of BSA molecules out of the films [78].

Innovative platform for tissue engineering was recently described by Azarfam and co-workers [79]. A composite based on gelatine/agarose and zeolite ZSM-5 was loaded with pomegranate peel extract. This extract is well known by its wound-healing activity through the promotion of haemostasis, anti-oxidative, immunomodulation, and antibacterial effects. Anti-atherosclerotic, anti-oxidative, and anti-parasitic activities are also associated with pomegranate extract [79]. HZSM-5 zeolite was explored as an ingredient on the composite formulation, wherein introducing zeolite into the hydrogel composites improves the integrity of the composite and decreases the swelling ratio and the pomegranate extract release behaviour, clearly showing the system to be useful as an MDDS [79].

### 3.2. Oral Formulations

Oral formulations represent the most common platform for drug delivery due to their convenience for patients. This administration route is simple and easy, non-invasive, and accessible for most people. Moreover, the wide variety of dosage forms that can be administered orally makes them a very versatile option. Additionally, it is the most permissive route in terms of the number and type of excipients that can be safely administered, such as zeolites.

Unmodified faujasite was recently found as an optimal candidate for the controlled release of drugs [80]. In particular, zero-order kinetic was found for the release of atenolol from faujasite in both gastric and intestinal simulated conditions. Moreover, the anti-tuberculostatic drug isoniazid was loaded into mordenite, faujasite-Y, and zeolite beta [81]. The resulting hybrid systems were addressed by full solid-state characterization together with molecular modelling calculations that were used to complement the study and understand the possible interactions, location, and orientation of the drug molecules in the zeolites. All the results demonstrated that isoniazid can be effectively adsorbed in the three zeolites. Nonetheless, the beta zeolite adsorbs more isoniazid than mordenite and faujasite-Y. Particularly, the atomistic calculations revealed that the drug is establishing interactions both in the zeolite channels and in the external surfaces and that it depends on the specific surface area of each mineral [81]. Additionally, the isoniazid loading capacity of faujasite-Y was optimized at pH 3, where a major part of the molecules are protonated in the NH_2_ group [82]. The zeolite was able to retain the drug without surface-precipitation phenomena. Even if the nanocomposite did not modify the release profile of isoniazid, faujasite-Y demonstrated that it was able to protect the drug from the acid media. Studies on the optimization of isoniazid adsorption by zeolite beta and the release behaviour were also reported [83]. These studies demonstrated that the maximum loading capacity is obtained at pH 6, and a minimum release of isoniazid was obtained in acidic conditions. Additionally, the results with both zeolites (faujasite-Y and beta) showed a lower thermal degradation of the drug retained in the zeolitic structure, a result that may be indicative of the protection that zeolite can give the drug against external conditions.

The aqueous solubility of two BCS II model drugs (nifedipine [84] and indomethacin [85]) was improved by their encapsulation in different zeolites. According to the results, almost 100% of encapsulation efficiency was obtained. Additionally, the in vitro dissolution study confirmed the usefulness of zeolites to improve the bioavailability of type II drugs (low-water-solubility drugs). Danazol is also a BCS II drug used to treat disorders such as endometriosis, fibrocystic breast disease, and to prevent hereditary angioedema. This drug has been loaded into faujasite by the so-called wetness method [86], producing the drug amorphization and, therefore, increasing its solubility. Moreover, authors reported remarkable stability of the system together with a controlled release of the active molecule.

*Acacia Catechu* extract is a catechin-rich extract with antimicrobial activity against *Bacillus subtilis*, *Staphylococcus aureus*, *Salmonella typhi*, *Escherichia coli*, *Pseudomonas aeruginosa,* and *Candida albicans*. The microencapsulation of the extract was performed in clinoptilolite. The results showed a satisfactory encapsulation in acid medium associated with electrostatic attraction forces between the protonated organic molecules of the extract and the oxygen atoms within the carrier framework. The in vitro release study showed that the microcarrier (clinoptilolite) was able to provide a sustained in vitro release of the *Acacia Catechu* extract [87].

One of the most important side effects of non-steroidal anti-inflammatory drugs (NSAIDs) is their aggressiveness towards the gastric mucosa, which could give rise to gastric ulcers. In this scenario, one of the possibilities would be to avoid the solubilisation and release of these molecules in the stomach, for which a MDDS is necessary. Ketoprofen was loaded into zeolite Y and into a zeolitic product obtained from a co-crystallization of zeolite X and zeolite A [88]. In this occasion, authors reported that less than 10% of the ketoprofen dose was released in the stomach. Moreover, the fact that the drug started to release significantly from pH 5 indicates that zeolites are perfect candidates to carry ketoprofen toward the final part of the gastrointestinal tract, where it can be used to treat inflammatory pathologies affecting this specific area. Diclofenac and piroxicam were also loaded into zeolite X and Y [6]. The in vitro dissolution studies demonstrated that the release of both drugs is pH dependent: minimum drug release in the stomach in comparison to the intestine. Authors ascribed this behaviour to the acidic nature of both drugs and to the establishment of electrostatic interactions between the drug and the zeolites. They stated that “the total surface charge become negative after the drugs enter into the intestine (pH = 6.8), and because of negative charge repulsion between the negatively charged surfaces of the zeolite with negatively charged drugs, drug released from the matrix is occurred but slowly” [6]. The anthelmintic drugs pyrantel pamoate and fenbendazole were loaded into zeolite Y in an attempt to obtain a slow drug release [89]. Results not only proved that zeolite Y was able to control the drug release of these anti-parasitic drugs, but also improved their efficacy and stability.

For oral dosage forms such as those previously mentioned, the biocompatibility of the formulation ingredients with the cells of the intestinal tract needs to be bear in mind. In this regard, the Caco-2 cells viability of zeolite Beta, ZSM-5, and NaX were evaluated by means of in vitro cytocompatibility tests. Results demonstrated that despite the intracellular accumulation of zeolite particles, no cytotoxicity was found [85], thus supporting the use of zeolites as safe excipients of oral dosage forms.

A different study reported the zeolites NaX, NaY, and HY as carriers of the poor water solubility clofazimine, an antibiotic used against a wide spectrum of Gram-positive bacteria, leprosy, and multidrug-resistant cancers [90]. In this study, the role of zeolites was compared with the mesoporous materials SBA-15, MCM-41, and Al-MCM-41. The silanol and OH groups, present in higher quantity on the mesoporous materials, created stronger interactions with clofamizine, while Na^+^, more frequent on zeolites structures, prevented stronger interactions with the silica framework. This resulted in the lower drug-loading capacity of zeolites NaX, NaY, and HY with respect to the mesoporous materials. Nonetheless, due to smaller zeolite pore diameter, stronger interactions between the drug molecule (clofazimine-clofazimine) are favoured [90].

Synthetic calcium zeolites A and X were tested as bisphosphonate carriers. In particular, sodium risedronate was selected as the active substance. This combination of ingredients was made in view of the affinity of bisphosphonates by calcium ions [91]. In order to guarantee a high amount of calcium ions in the zeolite structure, two cationic surfactants (dodecyl-trimethyl-ammonium chloride and benzyl-dimethyldo-decyl-ammonium chloride) were used during the synthesis of both zeolites. The authors reported that both zeolites were effective as carriers of risedronate, the mesoporosity, and the pore volume not exerting significant differences. The most remarkable differences were found in the release profile, zeolite A providing a long-lasting release and zeolite X providing a faster one.

Purified natural mordenite modified with Cu(II) as a coordinating agent was created to evaluate a new oral safe drug delivery material, for which the ibuprofen and meloxicam were used as drug models, and wherein the natural Cu(II)-zeolite has good physicochemical properties, with good retention for both ibuprofen and meloxicam, good cytocompatibility, and most outstanding release at pH 7 [92].

Table 1 summarises some of the unmodified natural and synthetic zeolites for several medicine applications. From zeolite as an active substance until zeolite as a drug carrier zeolite can accomplish a wide range of pharmaceutical applications.

#### 3.2.1. Organomodified Zeolites

Surfactants or tensioactives are versatile and varied molecules since they possess different lipophilic and non-lipophilic regions in their molecular structure. In the particular case of DDS and MDDS, surfactants have been used to functionalize different drug carriers including clay minerals and zeolites. In this process, surfactants act as bridges (intermediates) between the active substance and the carrier (in this particular case, the zeolite), thus improving the drug loading or influencing the drug release process. Figure 4 illustrate how this interaction can occur. For instance, the interaction of cephalexin with clinoptilolite depends on the zeolite-hexadecyltrimethyl ammonium functionalization [32]. Cephalexin interacted with pure clinoptilolite through simple surface adsorption, while the presence of the surfactant promoted cephalexin electrostatic and hydrophobic interactions. Another surfactant-modified natural clinoptilolite was loaded with diclofenac. NSAID drug usually administered in both oral and topical formulations [105]. In this study, cetylpyridinium chloride provided an effective loading and a sustained release of the drug throughout the gastrointestinal tract for up to 5 h [105]. The study carried out by Krajišnik et al. [106] was very similar. They modified clinoptilolite with both cetylpyridinium chloride and benzalkonium chloride prior to ibuprofen loading (another NSAID drug). Even though each surfactant provided different drug-loading capacity, the release profile was very similar: in both cases, approximately 30–40% *w*/*w* of ibuprofen was released within 8 h. The zeta potential of the functionalized clinoptilolites revealed that, despite the equivalent positive net charges of both surfactants and their similar molecular weights, structural differences and steric effects during the functionalization of clinoptilolite (among others), led to different arrangements of the surfactants and, therefore, to different drug-loading capacities. In the manuscript of Pasquino and co-authors [107], clinoptilolite was also modified with cetylpiridinium chloride and subsequently loaded with ibuprofen and diclofenac. Once again, the structure of the molecules determined the drug-loading capacity of the composite. In particular, diclofenac loading was higher than the maximum anion exchange capacity of the composite and with respect to ibuprofen. Authors related this difference to the fact that the “diclofenac molecular structure is more compact and the aromatic anions can insert more easily in between the surfactant molecules”. It is also worth highlighting the fact that a sustained release profile was found for both molecules again (up to 6 h). Continuing with cetylpyridinium chloride and clinoptilolite functionalization, the composite reported by [105] was successfully used by Serri et al. [108] to prepare an oral granulate. The formulation was able to provide a sustained diclofenac release for 9 h due to the ionic exchange process involved.

The functionalization of clinoptilolite zeolite with cetylpyridinium chloride (CC) was also reported by the study of Izzo and co-authors [109]. In addition to cetylpyridinium chloride, clinoptilolite was also functionalized with benzalkonium chloride, hexadecyltrimethylammonium chloride, and bromide. Due to the differential molecular structure, composition, and critical micelle concentration of surfactants, the combinations gave rise to different zeolite-surfactant formulations with distinct micellar structures. The Ibuprofen adsorption confirmed that the amount adsorbed was influenced by the surfactant type with the maximum and minimum loaded amount attained for the composite with hexadecyltrimethylammonium bromide and benzalkonium chloride, respectively. Two different diffusion processes were identified during the ibuprofen release: a first release associated with the drug adsorbed on the surface, and a second slower one related to the drug molecules trapped inside the hydrophobic chains of the surfactants [109].

#### 3.2.2. Zeolites and Polymers

Regarding zeolite surface modifications, polymers also play an important role for the formulation of oral dosage forms. Over the last decades, the use of polymers as pharmaceutical excipients in DDS and MDDS has significantly increased. This is due to their versatility, as they range from simple shells and coatings to more complex and comprehensible dosage forms with controlled or sustained drug release. In particular, polymer was highlighted because of their ability to improve solubility of certain drugs, swelling capacity, viscosity improvement, mucoadhesion, pH dependence… which are of great use in the design of oral dosage forms. As can be expected, the versatility of polymers together with zeolites makes the most of both excipients.

The most-used natural zeolite, clinoptilolite, has been combined with natural polymers to improve the physical properties of the latter. In particular, 3D polymer composites are susceptible to perform drug burst release due to their poor mechanical strength and high swelling ratio. A three-dimensional biocomposite based on chitosan was combined with clinoptilolite and loaded with indomethacin and diclofenac sodium, and the biocomposite was prepared by the green method of cryogelation, on which ice crystals are acting as a template preventing the formation of toxic byproducts or the use of severe chemical treatments for template extraction. For drug loading, the solvent evaporation technique was performed [110]. As expected by the nature of the polymer, the drug release was pHdependent, either with or without clinoptilolite. Nonetheless, the cumulative drug release was slower as higher the amount of clinoptilolite present in the composite (for both drugs).

Several publications report the formulation of nanocomposites based on zeolites and polymers for application as anticancer MDDS such as the synthesis and characterization of cross-linked poly(ethylene glycol) diacrylate nanogels containing natural zeolites which are loaded with 5-fluorouracil (5FU), an efficient chemotherapeutic drug for several species of cancers such as breast, rectal, and stomach cancers [111]. In this study, the zeolite–nanogels composite was obtained by an inverse mini-emulsion polymerization technique, a system that allowed obtaining a stable mixture of water-soluble polymer micelles composed of droplets of a polymer aqueous solution suspended by a mixture of surfactants and co-surfactants in a continuous organic medium [111].

Another example is the functionalization of Linde type L with poly-L-lysine to improve the cellular uptake of the composite and, therefore, improve the treatment efficacy. In this case, the zeolite surface was modified via attachment with NH_2_ groups, COOH groups, and coated with the polymer afterwards [100]. Polyethylene glycol was used as a surface modifying agent of zeolites Y and ZSM-5 by mixing, the nanocomposites obtained were studied as a delivery system for curcumin [112], and, finally, the coating of zeolite nanoparticles with a three-layer polymeric shell of chitosan-k-carrageenan-chitosan as a modified delivery system for verapamil [113].

Recently, the functionalization of zeolite-A with the biopolymer β-cyclodextrin to apply as carrier for levofloxacin (LVC) was reported and evidenced a significant improvement of the LVC loading capacity compared to zeolite-A as a single phase, a slow and continuous LVC release profile for 200 h, and a significant enhanced of the anti-inflammatory effect [114]. Table 2 summarize some applications of polymeric-zeolite composites on medicine and the method for drug loading.

### 3.3. Tissue Engineering

The scope of tissue engineering and regenerative medicine is the restoration of pathologically altered or injured tissues by transplantation of cells within supportive scaffolds and biomolecules. The so-called “scaffolds” or “construct” (that can be obtained by means of different techniques such as electrospinning or 3D printing and bioprinting) are intended to induce complete tissue regeneration in the wounded area. Therefore, the ingredients must be carefully chosen since they must act as analogues of the extracellular matrix and be fully biocompatible. Polymers and other macromolecules such as polysaccharides (among others) are optimal ingredients for the production of tissue engineering scaffolds. Nonetheless, these macromolecules fail to provide enough mechanical performance for certain kind of tissues such as bone, cartilage, ligaments, and tendons. At this juncture, inorganic ingredients such as graphene, hydroxyapatite, clay minerals, or zeolites come into play. In the particular case of zeolites, nanocomposite scaffolds have emerged as a new approach to bone regeneration by tissue engineering methods, offering an alternative to the traditional treatment, as shown in Table 3. The presence of zeolites within polymeric scaffolds award them with additional properties, mainly in terms of mechanical reinforcement.

In this context, nanocomposites scaffolds made of polylactic-*co*-glycolic acid (PLGA)/zeolite were prepared by electrospinning [120]. A sodium-rich zeolite was synthetized and added to the PLGA obtained for this purpose, whose main scope was to improve the physical, mechanical, and biological properties of the scaffolds, intended for bone regeneration and tissue engineering. Three different zeolite concentrations were tested (3, 7, and 10% wt.), the results proving that the biodegradation of the resultant nanocomposite can be controlled by the addition of the mineral in a directly proportional manner. Regarding cellular viability and biocompatibility, the osteosarcoma MG63 cell line effectively adhered to the nanocomposites. In terms of proliferation, the results revealed that the nanocomposite with 7% of zeolite was the most suitable for bone tissue engineering. Continuing with bone tissue engineering, a poly(ε-caprolactone)-poly(ethyleneglycol)-poly(ε-caprolactone) (PCL-PEG-PCL) triblock copolymer, with recognisable usefulness for this particular scope, was combined with clinoptilolite [115]. The clinoptilolite/PCL-PEG-PCL scaffolds showed higher interconnected porous structure and higher mechanical resistance in comparison with pure PCL-PEG-PCL scaffolds. Likewise, the presence of clinoptilolite improved water adsorption capacity, which is of crucial importance to facilitate the diffusion of proteins and solutes within the structure. The in vivo and in vitro evaluation of the clinoptilolite/PCL-PEG-PCL composite scaffolds showed an efficient tissue regeneration with the highest quality of bone union, cortex development and bone-scaffold interaction at the defect site. These promising results proved the useful potential of clinoptilolite (and, therefore, of zeolites in general) as mechanical reinforcement of polymer-based scaffolds intended for tissue engineering. The clinoptilolite/PCL-PEG-PCL composite scaffolds were evaluated in vivo and the presence of zeolite on scaffold composition improving bone regeneration and promoting “mechanical, physical, and biological properties of polymer-based scaffolds in a more economical, easy-to-handle, and reproducible approach” [121] was proven. The effect of zeolite on cell viability, proliferation, osteo and odonto-genic differentiation, and ultimate mineralization of human dental pulp stem cells (hDPSCs) cultured on PCL-PEG-PCL nanofibers was also reported by Alipour and co-workers [122]. In fact, the highest cell viability, proliferation, and differentiation were obtained PCL-PEG-PCL/Zeolite.

ZSM-5 zeolite, beta three-calcium phosphate (βTCP), and gelatine have been used to prepare a biocompatible scaffold for bone regeneration [123]. A scaffold without zeolite (βZG 0) and with different amounts of zeolite (βZG1 and βZG2) were synthesized by freeze-drying and compared. The results showed that increasing the percentage of ZSM-5 increases the compressive strength of the composite, something ascribed to the scaffold porosity, an important property for bone generation. The βZG 2 scaffold showed better results in cell viability, although the amount of zeolite in this sample was considered excessive and detrimental for the mechanical properties: reduced flexibility and increased fragility of the composite. Therefore, the βTCP/ZSM-5/Gelatin scaffold can be an appropriate candidate for medical application in bone regeneration [123]. Zeolite A nanocrystals were combined with chitosan to produce a solid, dry scaffold through freeze-drying. The effects of the resultant nanocomposite scaffolds on cell attachment, survival, and proliferation of human mesenchymal stem cell line were evaluated. The results indicated that the scaffolds prepared with 0.5% wt zeolite A provided better cellular attachment and survival with respect to its counterparts containing 1% and 2% wt of zeolite A [119]. In wound-healing and tissue regeneration, the prophylaxis of infections is of great importance, especially when the treatment deals with implant of any construct of device. In this respect, synthetic, Ag-loaded VPI-7 zincosilicate zeolite scaffolds (Ag-3DPZS) were fabricated via 3D printing, aiming to obtain antimicrobial, tailor-made, implantable three-dimensional constructs for bone tissue regeneration. The Ag-3DPZS scaffold reported to be effective against a broad-spectrum of microorganisms, among which *S. aureus* and *E. coli* can be mentioned. More importantly, the scaffold proved to be non-cytotoxic, which is of crucial importance for the healing and regeneration of tissue [124].

Osteosarcoma is one bone tumor which causes damage to nearby tissues. Recently, zeolite faujasite Y and polycaprolactone nanocomposite were fabricated as carriers for cisplatin, a chemotherapeutic drug used for osteosarcoma treatment as an alternative to develop an in situ drug delivery system which also functions as a scaffold for bone regeneration, as demonstrated at Figure 5 [125]. Based on results of mechanical properties, biodegradability, in vitro bioactivity, and drug release, among other things, the authors observed that the zeolite improved the scaffold mechanical characteristics, providing a pH-sensitive and drug-sustained release, therefore being strong candidates for regeneration of the bone-cancer-affected tissues.

### 3.4. Zeolites for Vaccines and Other Parenteral Dosage Forms

It is no secret that vaccinations have had a huge impact in human health worldwide, being the most life-saving advance in medicine’s history. Due to vaccines, it’s been possible to prevent and even eradicate numerous diseases with high mortality, SARS-CoV-2 being the best-known current example.

One of the first vaccine–zeolite relationships dates back to 1981 [126]. In this study, a natural zeolite was used to inactivate *Trypanosoma gambiense* parasites by keeping them in contact at 4 °C for 3 days. The mechanism of parasite inactivation was based on the remarkable CEC of these minerals. The natural zeolite used in this study was demonstrated as useful in the vaccine production. In fact, rabbits injected with the inactivated *T. gambiense* vaccine were completely protected after 1–2 weeks after the immunization, reporting effective parasite agglutination [126].

Natural microparticles of clinoptilolite (commercialized under Cliptox^TM^) have proved to be an effective vaccine adjuvant. The foot and mouth vaccine consists of an inactivated virus prepared by using Cliptox^TM^ as an adjuvant [127]. The murine model results demonstrated that vaccines including clinoptilolite provided better protection with respect to those without the zeolite adjuvant due to an increased immune response. Consequently, clinoptilolite can be considered as immunomodulation ingredient. In another study, a DNA vaccine against bovine herpesvirus-1 was prepared by Langellotti et al. [128], and the role of different adjuvants evaluated. In this study, clinoptilolite produced the best results among all the adjuvants under study since it was able to induce specific humoral and cellular immune responses in mice.

The participation of inorganic ingredients for the formulation of intravenous dosage forms is well known. In fact, magnetic nanoparticles have been widely explored for imaging, diagnosis, and drug–organ–target treatments, enhancing the treatment effectiveness and reducing the distribution of the drug throughout the organism (and subsequently minimizing side effects) [129,130]. In this regard, zeolites have been proposed as support for magnetism. In fact, the magnetization of zeolite NaX for local anticancer treatment has proven to possess great potential, with good results in cancerous cells mortality [118]. By the same token, the zeolite beta has been explored as a magnetic platform after its modification with poly(di-allyl-dimethyl ammonium) chloride [130]. This study explores and optimizes the amount of magnetite in the zeolite structure and uses thiamine hydrochloride (vitamin B1) as a drug model. They concluded that the most optimal system is obtained with a concentration of 10–12% *w*/*w* of Fe_3_O_4_.

#### Theranostic Systems

Theranostic systems can be defined as those combining diagnosis with treatment in the same formulation/system. In fact, the word “theranostic” is basically an acronym for the words “therapeutic” and “diagnostic”, and it is a growing field of study. The majority of theranostic systems are administered by parenteral route.

Some studies have revealed the currently active role of zeolites in this field. For instance, NaY zeolite loaded with magnetic nanoparticles was evaluated as a T_2_-MRI contrast enhancer in magnetic resonance imaging (MRI) [131]. The suitability of these NaY-magnetic nanoparticles has been assessed by their biocompatibility and non-toxicity, which were accomplished even at the highest concentrations. Moreover, an effective cellular internalization of the particles by tumour cells guaranteed a significant accumulation in the tumour site, enough to be detected as a darker contrast in the MRI image.

Magnetic nano and micro-particles of clinoptilolite (MCZ) were studied as controlled DDS, imaging, and local heating in biological systems [132]. The magnetic nano/micro-particles were prepared by incorporating iron oxide Fe_3_O_4_ into clinoptilolite matrix to create the magnetic properties. In addition, rhodamine B (temperature sensitive dye, for diagnosis), sulfonated aluminium phthalocyanine (a photosensitizer widely used in photodynamic cancers therapy), and hypericin (a powerful photosensitizer with pronounced tumor-localizing properties) were loaded as actives. In particular, the release of hypericin was investigated, showing two different rates depending on the location of hypericin molecules: the faster release can be ascribed to molecules located on the surface of the MCZ clusters, whereas the slower release is probably associated to the hypericin molecules inside the MCZ clusters. Furthermore, the MCZ particles were effective as local hyperthermia inducers according to the in vitro results obtained with a chicken organ. It is well-known that the induction of 39–45 °C improves the therapeutic outcome in several tumours. In particular, 39.7 °C were reached within 22 min, thus indicating the effective hyperthermic activity of the MCZ system [132].

### 3.5. Anticancer

Cancer is, in general, a complex disease with poor prognosis and high morbidity. The majority of cancers are generally treated with traditional, aggressive, and unspecific chemotherapies, which are characterized by numerous side effects. In this context, the combination of chemotherapy and nanoparticle-based DDS and MDDS have shown great potential for progress in cancer therapy [133,134]. Natural clinoptilolite has been combined with graphene oxide and loaded with doxorubicin [94]. One of the main problems of doxorubicin is cancer drug resistance. The clinoptilolite/graphene oxide composite demonstrated a high doxorubicin-loading capacity and increased cytotoxicity against in vitro lung cancer cells, thus demonstrating improved effectiveness. The efficiency of MFI-type borosilicate zeolites as DDS for doxorubicin was also reported [104]. In this system, it is worth mentioning that the drug-loading efficiency was greatly influenced by the borosilicate particle size, the organization of the zeolite nanocrystals, and the external surface area of MFI borosilicate. Moreover, the in vitro release of doxorubicin from the MFI borosilicate particles proved to be pH dependent, with a higher drug release rate in low pH environments [104]. This pH-dependent release of MDDS has already been mentioned elsewhere in this manuscript as a good strategy to control the release of certain drugs along the gastrointestinal tract. Interestingly, for cancer therapy, this feature is also of great interest since the extracellular pH around the tumour lesion has acidic values. Therefore, a higher amount of doxorubicin can be released around the tumour, concentrating the drug in the specific action site and, subsequently, potentially reducing chemotherapy side effects. Recently, doxorubicin was used as a model drug for the development of a MDDs, with sustained release and real-time drug release monitoring functions in which the desilicated ZSM-5 zeolite with mesoporosity acted as drug carrier and NaYF4: Yb^3+^/Tm^3+^ nanocrystals functioned as monitoring probe [135]. Following a ship-in-a bottle route, NaYF4: Yb^3+^/Tm^3+^ nanocrystal was synthetized by impregnation in ZSM-5 previously desilicated and the DOX loaded inside this composite, wherein this composite made possible the release monitoring, and also has a good biocompatibility.

Zeolite beta was evaluated as a mitoxantrone carrier [96]. Mitoxantrone molecules were strongly adsorbed at the microporous surfaces of the zeolite. According to the solid-state characterization, the β-hydroxyketone functional moiety of the drug interacts with the structural Al of the zeolite. The in vitro culture studies revealed that the cytotoxic activity of mitoxantrone was maintained, even if the drug release from beta zeolite was slow. These results lead authors to state that the anticancer effect would occur once the zeolite nanoparticle is internalized by cancer cells [96].

Synthetic faujasite and linde type A were evaluated as α-cyano-4-hydroxycinnamic acid carriers [98]. The resultant DDSs were able to carry and release the anticancer drug in therapeutic doses. What is more, the encapsulation of the anticancer drug in these two zeolites proved to improve the efficacy in comparison with pure α-cyano-4-hydroxycinnamic acid.

More recently, faujasite zeolite was studied for the first time on X and Y forms as a carrier for 6-mercaptopurine (MERC), a chemotherapeutic drug used to treat various diseases such as blood cancer, inflammatory bowel disease, or Crohn’s disease [136]. The current methods of administering this drug present poor bioavailability, a short half-life in plasma, and provoke side effects such as bone marrow suppression and hepatotoxicity. To enable the adsorption interaction between the zeolite and the drug, faujasite had an ion exchange with Zn^2+^ ions, making possible the formation of complexes with sulphur and nitrogen atoms from MERC (Figure 6a). The results proved the adsorption of the drug without the evidence of drug precipitation and zeolite aggregation Figure 6b, and 78% release of the drug following a controlled manner. An important aspect in the use of zeolites for this purpose is the non-destruction and aggregation of the particles and the non-precipitation of drugs. As seen in Figure 6b, significant changes in the morphology of the zeolite crystal are not observed after the adsorption step, a behavior that collaborates to prove the effectiveness of these zeolites as carriers.

NaX-faujasite and zeolite beta were loaded with 5-FU [28]. NaX-faujasite provided a higher loading capacity and faster release (total drug release in 10 min). On the other hand, zeolite beta showed a multi-step release profile prolonged with time. By means of molecular dynamics, these differences were ascribed to stronger van der Waals interactions between 5-FU and beta zeolite in comparison with NaX-faujasite. Regarding effectiveness, the effect of zeolitic particles on the viability of Caco-2 monolayers showed that the NaX-faujasite particles cause a higher reduction in cell viability with respect to beta particles. In another study, 5-FU was again loaded into NaX together with ZSM-5 and zeolite A [42]. All of these zeolites were previously micronized in order to normalize their particle size and maximize their specific surface area. The drug release profile was pH dependent, being significantly higher at pH 1.6 than at pH 5. In this case, the faster drug release found at lower pH was related to the degradation of the zeolites, which was in agreement with the release of Na^+^, Al^+,^ and Si_4_^+^ in both environments. Due to this zeolite degradation, authors tested the possible cytotoxicity of the composites against intestinal Caco-2 cells in an attempt to discern if the cellular mortality was related to the zeolites, the drug, or both. Results allowed the discarding of any “harmful” effect of the composites towards other cells, finding full cellular biocompatibility [42]. Later, Golubeva and co-authors [137] studied 5-FU loading onto four different aluminosilicates: saponite, montmorillonite, halloysite nanotubes, and beta zeolite. All these ingredients possessed the same chemical composition but with different structures and morphologies. In addition, different solvent mediums were evaluated during drug loading (ethanol, acetone, neutral, alkaline, and acidic medium). Results demonstrated that acetone and aqueous alkaline medium were the most optimal for 5-FU loading onto aluminosilicates. Moreover, beta zeolite performed the best 5-FU adsorption (either in acetone or aqueous alkaline medium). In a more recent study, 5-FU was adsorbed on the natural clinoptilolite zeolite functionalized by Na^+^ ions (G.Na^+^/Clino), and for this functionalization a green tea extract was utilized, a process named by the authors as “green functionalization” [138]. They observed a higher loading capacity of G.Na^+^/Clino and a continuous and longer release profile that took about 150 h. Furthermore, the cytotoxicity study for the hybrid material showed a significant increase for the impact of the loaded drug on cancer cells.

### Zeolite Nanocomposites for Anticancer Drug Delivery

Although there are several studies related to the use of hybrid materials composed of zeolite and the drug of interest, sometimes only the zeolite–drug interaction is not enough to meet the technological need in the development of a given drug. For this reason, the study of nanocomposites on the advantages associated with the use of zeolites with other elements such as biocompatible polymers, for example, is also an alternative for the development of more effective drugs. An example of this is the study carried out by Spatarelu et al. [111], wherein 5-FU was loaded on an unspecified natural zeolite, and after drug loading, the natural zeolite was included in a poly(ethylene glycol) diacrylate nanogel composite by using an inverse mini-emulsion technique. Both the pure natural zeolite (loaded with the anticancer drug) and the polymer/zeolite nanocomposite were compared. In terms of drug release, the zeolite system provided a faster 5-FU release. On the other hand, the polymer/zeolite composite achieved a total release of 40–45% of drug with a lower burst release, thus demonstrating its potential as a MDDS.

Linde type L has also been used as 5-FU carrier [100]. The in vitro studies revealed that the anticancer activity was potentiated by the zeolite and that the cellular internalization of the systems occurred by caveolin-mediated endocytic mechanism. Therefore, authors tested the internalization of the composite after functionalization of the zeolite with NH_2_, COOH groups, and with poly-L-lysine in order to study whether surface modification enhances the cellular uptake. As expected, positively charged zeolites showed higher cellular internalization with respect to negatively charged and the non-functionalized Lynde type L/5-FU composites, thus improving tumour reduction. What is more, the increase in cellular uptake did not imply higher toxicity to non-tumoral cells.

Nanoparticles of ZSM-5/KIT-6 and ZSM-5/SBA-15 were studied as drug delivery systems for verapamil. The final scope was to investigate the intracellular traffic of doxorubicin [113]. The composite zeolite/mesoporous system combines the zeolite interconnected networks with mesopores, being able to offer good possibilities of functionalization and drug interactions. In this study, the nanoparticles were also modified with -(CH_2_)_3_SO_3_H and -(CH_2_)_3_NHCO(CH_2_)_2_COOH groups and coated with a three-layer polymeric shell composed of chitosan-k-carrageenan-chitosan, resulting in a modified in vitro release of verapamil for a period of 24 h. The drug release was influenced by the nanocarrier structure, the functional groups, and the presence of the polymer complex [113].

Curcumin is a natural molecule with a wide variety of activities, among which the anticancer activity is highlightecd. Curcumin was loaded in zeolite Y, ZSM-5 [112]. Zeolite 5A (the calcium-exchanged form of zeolite Linde type A) has also been proposed as curcumin carrier [98]. This zeolite-curcumin combinations are established through hydrogen bonds, as schematically represented on Figure 7 [98]. Zeolite Y and ZSM-5 were modified with polyethylene glycol, and the loading and release properties were compared with the corresponding unmodified systems [112]. The surface modification with polyethylene glycol reduced the curcumin loading efficiency in both zeolite Y and ZMS-5. Nonetheless, the release of the drug from the modified nanocomposites was higher with respect to pure zeolites, which means that the presence of the polymer improved the water solubility of the drug and, therefore, its release from the dosage form. In the study reported with zeolite 5A [112], not forgetting the differences in the methodology, the loading capacity was slower than in unmodified zeolite Y and ZSM-5 [98]. This could be ascribed to the smaller channel size of zeolite Linde Type A with respect to zeolite Y and ZSM-5 [139]. Nanosized ZSM-5 functionalized with β-cyclodextrin (β-CD) has been also investigated as a drug delivery system for curcumin [99]. The ZSM-5/β-CD nanocomposite showed that curcumin release was in acidic conditions rather than in neutral environments. As previously mentioned, the extracellular environment of tumours has acidic pH values. Once again, this pH-dependent release is of great use to concentrate the chemotherapeutic agent around the cancerous tissue, which would increase effectiveness and reduce side effects.

A magnetic nanozeolite NaX has been recently proposed as a site-specific anticancer drug carrier [118]. According to the authors, the magnetic NaX particles are easily transported by the bloodstream, though zeolite particle agglomeration could jeopardize the safety and effectiveness of the formulation. To solve this problem, they proposed the incorporation of the magnetic NaX nanozeolite within a polymeric matrix made of poly(lactic acid) (PLA) and chitosan. The resultant composite consists of magnetic NaX/PLA/chitosan nanofibers loaded with doxorubicin. This formulation reported a doxorubicin loading efficiency of >90%. Moreover, a doxorubicin sustained release was reported under the influence of a magnetic field, with 82% of H1355 carcinoma cell death, which revealed the system as highly effective for local chemotherapy treatments [118]. Other zeolite/polymer nanocomposites were also reported as successful anticancer MDDS. In particular, zeolite Y and ZSM-5 were explored as paclitaxel anticancer drug carriers in combination with poly lactic co-glycolic acid and chitosan [116]. Both zeolites performed a similar controlled drug release, which was maintained for more than 700 h at pH 7.4 and pH 5.5.

### 3.6. Miscellaneous Dosage Forms

In this section, all the research works that do not specify the dosage form designed or those in which the formulation could be administered by more than one route at the same time are included. Moreover, treatments such as haemodialysis, in which no dosage forms are used, have also been included in this section in view of their significance.

The remarkable adsorption capacity of zeolites is not only useful for the adsorption and controlled release of drugs, but also toxins. During haemodialysis, uremic toxins must be eliminated from the blood with an exogenous filtration system basically formed by a mesoporous membrane [29]. In this regard, different zeolites, linde type A (LTA), stilbite (STI), silicalite (MFI), mordenite (MOR), and faujasite (FAU), were proposed and studied as dialysis membrane ingredients and compared with activated carbon. The adsorption capacity of uremic toxins by each zeolite was evaluated and compared in terms of channel size, acidity, hydrophobicity, compensating cations, and grain size. The results were very promising since the adsorption process not only was effective but also specific: according to Wernert et al., “MFI, MOR (Si/Al = 10) and K-STI are very efficient with indoxyl sulphate at 37 °C, while creatinine can selectively be adsorbed by the MOR already mentioned. At 37 °C Ca-STI, K-STI and Na-STI are all three selective regarding the adsorption of uric acid while Na-STI is with urea”. Therefore, these results demonstrated that it is possible to modify and combine different zeolites to synthesize an effective dialysis membrane with remarkable adsorption capacity. In the same line, para-cresol (p-cresol) is a protein-bound solute which is not efficiently removed by dialysis membranes due to its low diffusion capacity. Therefore, silicalite (hydrophobic zeolite) was evaluated as an adsorption platform for this molecule [140]. This study concluded that to increase the capacity of purification of p-cresol, it should be possible to add a dialysis element containing silicalite with a high affinity by p-cresol. An elimination of 90% of p-cresol in 2 min is considered “high affinity”. Recently, the imprinting technology was used to produce a zeolite-Y having a pore size which was in accordance with p-cresol molecule [30]. This modification increased the p-cresol adsorption capacity to 2.5 and 3.5 times higher with respect to other non-imprinted zeolites (conventional synthesis and commercial zeolite-Y). Electrospinning has also been proposed as a technique to combine polymers and zeolites in order to develop haemodialysis membranes to effectively remove toxins from bloodstream [141]. Poly(ethylene-co-vinyl) alcohol was chosen due to is blood biocompatibility, while different zeolites (mordenite, ferrierite, ZSM-5, and beta, with different Si/Al ratios) were screened. The zeolite beta with Si/Al ratio = 37 showed the highest creatinine adsorption capacity. Authors stated that “although the barrier properties of the EVOH matrix lowered the creatinine adsorption capacity of the zeolites in the fiber when compared with adsorption to free zeolites, their adsorption capacity was still 67% of the free zeolites”.

Methylprednisolone hemisuccinate was loaded in a wide range of synthetic zeolites in order to study the loading capacity and drug release [27]. Authors aimed to obtain a release profile to treat rhinosinusitis (sustained release). The in vitro release studies were performed in simulated blood/tissues and tumour environments. Two impregnation methods were used to load the corticoid molecule into different 2D, 3D mesoporous materials (MCM-41, SBA-15, expanded SBA-15, FDU-12, and SBA-16) and two hierarchical zeolites (h-ZSM-5 and h-BETA). The results of h-BETA and SBA-16 were the most favourable for rhinosinusitis treatment since both platforms performed the most sustained release over time, as shown in the Figure 8.

Lysozyme plays a fundamental role as a natural prophylactic enzyme against bacterial infections in the human body. Based on this enzyme, implantable devices loaded with lysozyme are envisaged as a useful strategy to reduce infections, inflammation, and the eventual need for implant replacement. In this context, lysozyme sorption by pure-silica zeolite MFI films was studied. The influence of film orientation, incubation time, and incubation volume on the amount of adsorbed lysozyme were studied [103]. MFI films are able to adsorb lysozyme effectively. Maximum adsorption capacity can be obtained by increasing MFI-lysozyme contact time or the lysozyme concentration. On the other hand, the adsorption process is not influenced by the MFI crystal orientation (randomly or b-oriented) on the films.

Two natural Chilean zeolites (clinoptilolite and mordenite) were micronized to obtain zeolite nanoparticles. Immediately afterwards, they were doped with Cu^2+^ to award antimicrobial activity. The microbiological tests showed that both zeolites doped with copper salt ions were effective against *Escherichia coli* and *Staphylococcus aureus*, highlighting the potential application of natural Chilean zeolites as a copper-based antibacterial systems [93]. Although no administration route is specified in the original manuscript, this could be a proper formulation for topical administration either to prevent or to treat infected chronic wounds in the form of semisolid formulations or wound dressings (prepared, for example, through electrospinning or 3D printing). In the same line, a nanostructured zeolite X ion-exchanged with zinc, copper, and iron was a great host for metal ion-based antimicrobials [142]. The rapid metal ion release from zeolite X provided a fast and effective antimicrobial activity against *Staphylococcus aureus.* Zeolite X, zeolite Beta, zeolite Rho, and paulingite have been used as silver carriers [101,102]. Kwakye-Awuah et al. [101] reported that Ag-zeolite X was effective against *E. coli, P. aeruginosa* and *S. aureus*, even after the retrieval of the system from previous cultures. That is, the composite reported to be recyclable for further uses. Golubeva and co-workers [102] introduced silver in the mineral platforms as nanoparticles and clusters in order to evaluate and compare their antimicrobial and antitumoral activity. Zeolite Rho showed the highest antimicrobial activity with respect to zeolite Beta and paulingite. Due to its structure, Rho zeolite stabilizes Ag_8_ nanoparticles and clusters, while paulingite and Beta zeolites stabilize nanoparticles and clusters of Ag^+4^ and Ag_8_. Nonetheless, Rho zeolite also showed the highest cytotoxicity to both tumoral and normal eukaryotic cells. Thus, regarding the antimicrobial effect, Rho zeolite would perform the better activity, while paulingite would be the optimal carrier when looking for an antitumoral effect since it showed the highest activity with the lowest cytotoxicity toward normal cells [102].

Eucalyptus essential oil, with antibacterial, anti-inflammatory, and antipyretic activity, has been loaded into a zeolite Y modified with β-cyclodextrin [143]. The cyclodextrin-zeolite modification has been performed by means of three different crosslinkers (citric acid, succinic acid, and adipic acid). The less convenient crosslinker demonstrated to be succinic acid since it performed the lowest grafting yield and released gaseous essential oil less efficiently. Despite these disadvantages, this study demonstrated that by choosing the proper crosslinker, the cyclodextrin-zeolite Y system had a remarkable potential as essential oil carrier and control release.

A zeolitic, pH-responsive drug release system was prepared by coating zeolite Y with tannic acid [144]. Metronidazole, an anti-protozoan drug commonly used for the treatment of infectious diseases caused by anaerobic bacteria and protozoans, was loaded into the modified zeolite. A controlled and pH-sensitive release behaviour was provided by this zeolite Y/tannic acid platform. Authors defined two differential roles of each ingredient: zeolite pores harboured, carried, and protected metronidazole, whereas tannic acid acted as a barrier that controlled the drug release profile.

The natural zeolite B was used as a mechanical reinforcement of a polyethylene glycol acrylate hydrogel. The composite was loaded with Rose Canina extract, which possesses antioxidant and anticancer activity [117]. According to the experiments, all the ingredients established chemical interactions between them, thus improving the hydrogel performances. In fact, the zeolite participated in the crosslinking of the hydrogel, together with the rose hip extract. In addition, the presence of zeolite type B reduced the burst release of the extract.

ZSM-55 is a synthetic zeolite with a layered structure, which has recently attracted the attention as a possible drug carrier [145]. In this study, the role of ZSM-55 as drug delivery system was tested with two different molecules: piracetam (high aqueous solubility) and ciprofloxacin (low aqueous solubility), both of them loaded into ZSM-55 by the traditional intercalation solution method. Both drugs were loaded into detemplated ZSM-55 and pillared ZSM-55. “Detemplated ZMS-55” refers to the resultant layered zeolite obtained after its synthesis in the presence of unspecified organic structure directing agents (SDA), with the different layers connected with each other by weak hydrogen bonds (Figure 9); pillared ZSM-55 refers to the layered zeolite that has been subjected to “pillarization”, thus increasing the distance between the layers (Figure 9). In this case, the drug molecule was more determinant for the loading capacity than the zeolite (whether detemplated or pillared). In fact, the lower ciprofloxacin loading capacity by both systems was ascribed to the low water solubility and molecule. On the other hand, piracetam was successfully intercalated in both system, though with higher amount in the pillared sample. Finally, the “slowed down release of piracetam” was ascribed to stronger drug-carrier interactions and “better solubility and faster release of ciprofloxacin was attributed to amorphization of the drug, formation of its protonated form and weaker interactions with the zeolite than with other drug molecules” [145].

### 3.7. Computational Studies in Zeolite-Drug Interactions

The development of new carriers that allow the formulation of more efficient medicines is a field that has been exhaustively considered by the scientific community, exploring different types of materials, among them clays, polymers, carbon nanotubes, and zeolites and their modifications as widely discussed in this review. However, understanding how a carrier and the drug can interact is possible through exhaustive experimental studies, which are not always able to provide the necessary information for understanding the mechanism. In this context, computational studies act as a way of predicting the connections and dynamics between the carrier and the drug, thus making it possible to understand the mechanisms involved in both drug adsorption and release. In fact, Fatouros et al. [95] proved that molecular dynamics are very useful to screen drug–zeolite composites. In particular, they used theophylline and salbutamol as model drugs and evaluated the different theoretical drug-loading procedures. Even if both drugs had similar widths in their lowest energy conformations, the molecular dynamic calculations showed that salbutamol was able to move through the zeolite framework, whereas theophylline was not. This difference was ascribed to the flexibility of salbutamol, theophylline being a more rigid molecule. “These results suggest that any experimentally observed release for theophylline is likely to be controlled only by the surface of the zeolite since molecules are unable to move in or out of the pore channel system” [95]. The low-water-solubility probucol (antilipidemic drug) was loaded into three different zeolites by the wetness impregnation method [21]. The zeolites used were Beta, MFI (silicalite), and zeolite NaX. This study was complemented with molecular dynamic simulations. Due to the drug-loading technique, drug amorphization was observed for all the zeolitic platforms. Probucol molecules placed both in the external surface and inside the micropores. Regarding the release process, in vitro studies coincided with the molecular dynamic simulations: as higher the aluminium content, the lower the probucol release. Molecular modelling studies were performed to better understand the interactions between the isoniazid molecule and the zeolites beta, mordenite, and faujasite [81,82,83]. On this occasion, the molecular modelling studies were in agreement with the experimental data, thus offering a good theoretical approximation [81,82,83]. Molecular modelling by quantum calculations at Hartree−Fock and density functional theory were conducted in the study performed by Datt et al. [146] in order to understand the intermolecular interactions between the zeolite HY zeolite and the aspirin.

## 4. Conclusions

The development of new materials that can act efficiently in the improvement of drug formulations and treatments is a big challenge since it is a multidisciplinary area that covers several fields of knowledge. When approaching the possibilities of zeolite applications in this scenario, its versatility is evident, being able to act from active elements to as a function of carrying drugs in a specific location. It is also evident that many parameters must be considered when it comes to the use of zeolites for drug formulations and treatments, such as porosity and their composition in relation to possible isomorphic substitutions and thus the existence of exchangeable cations, among other things. Moreover, the large number of possible zeolites that can be applied between natural and synthetic and, for each case, the possibility of functionalization to meet the specific needs of the desired pharmaceutical applicability makes it necessary to study case by case, considering both the drug and/or purpose, the type of zeolite, and which parameters can be improved for a more viable performance. Through this bibliographical study, it is possible to highlight some points:−The application of natural or synthetic zeolites as active substances is something that already exists in our society. For this, the property of cationic exchange is what enables its commercialization, acting as regular agents of stomach acidity, adsorbent of contaminants, and as accelerators in the healing of wounds.−The possibility of inserting elements such as Zn and Ag, also associated with the retention of drugs, opens a range of possible applications as topical treatments.−Studies of oral administration formulations show a great interest in the use of zeolites. Additionally, in this segment, they can act protecting the drug from degradation and release in unwanted areas, as well as enabling the dissolution of those hydrophobic substances. This makes orally administered drugs for anticancer treatment widely studied and associated with zeolites since the treatment against cancer in many cases generates more severe side effects, which may be associated with immediate release or undesired location.−When it comes to drugs that naturally would not have a direct interaction with the possibilities of easily accessible and synthesized zeolites, the functionalization of their surface, either with surfactants, metals, or with polymers, is a strategy already adopted, working as well as bridge of interaction between the inorganic and organic element.−The use of computational tools that can predict the possibility of interaction between a drug and zeolites is an important strategy to enable both the reduction in experimental studies and the understanding of the adsorption mechanisms involved.

With this, we can predict that, due to their functionalization versatility regarding composition and porosity via composites and hybrids, zeolites may in the coming years revolutionize the way medicines are produced, opening up a new possibility of materials, in addition to well-established clays and polymers, to be explored as tools in the production of more effective products and treatments.

## Figures and Tables

**Figure 1 pharmaceutics-15-01352-f001:**
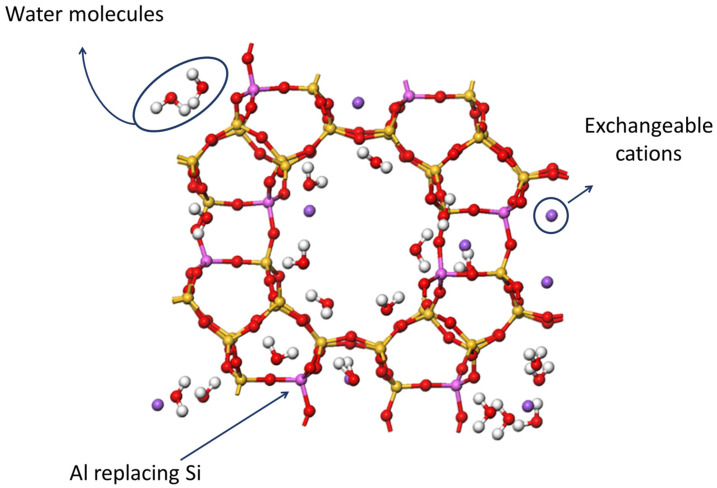
Schematic illustration of the structure and components of a zeolite. Original reproduction.

**Figure 3 pharmaceutics-15-01352-f003:**
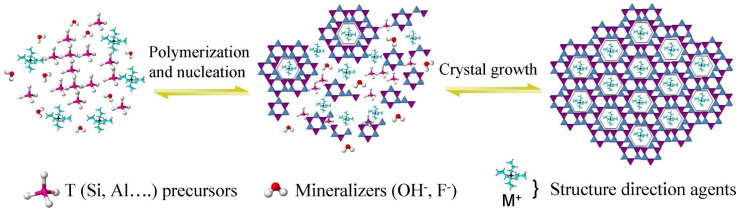
Simplified scheme for the synthesis of a zeolite. Reproduced with permission from reference [37] published by Elsevier, 2013.

**Figure 4 pharmaceutics-15-01352-f004:**
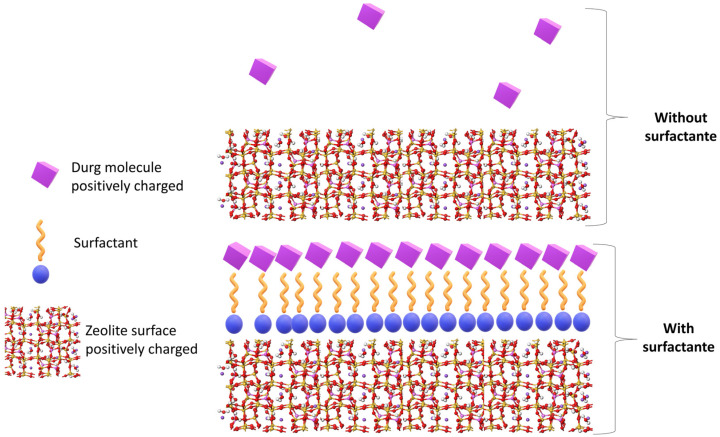
Zeolite surface modification with surfactant and interaction with drugs molecules. Original reproduction.

**Figure 5 pharmaceutics-15-01352-f005:**
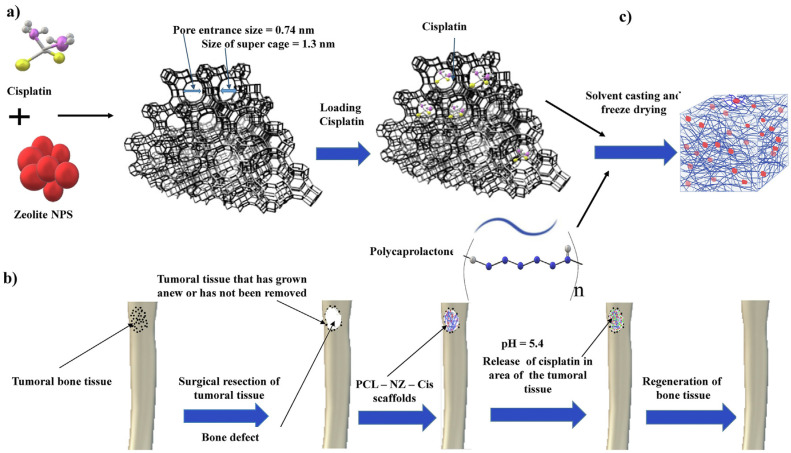
The schematic illustration of the nanocomposite synthesis acting as drug release and scaffold for treatment of osteosarcoma: (**a**) drug load inside zeolite nanoparticles followed by (**c**) the formulation of nanocomposite by addition of polycaprolactone; (**b**) implantation of nanoparticles scaffolds in bone defect and its performance. Reproduced with permission from reference [125] under creative commons license, published by Elsevier BV, 2022.

**Figure 6 pharmaceutics-15-01352-f006:**
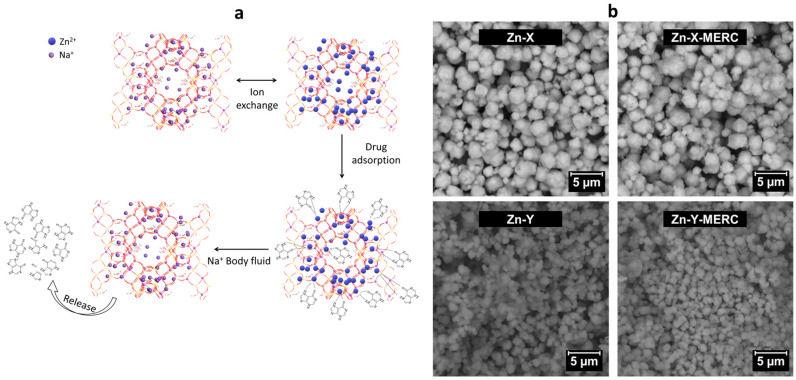
(**a**) Influence of ion exchange on drug adsorption of MERC on faujasite zeolite; (**b**) SEM images for zinc zeolite X and Y before (Zn-X and Zn-Y) and after sorption of MERC (Zn-X-MERC and Zn-Y-MERC). Modified with permission from [136] under creative commons license, published by Elsevier, 2022.

**Figure 7 pharmaceutics-15-01352-f007:**
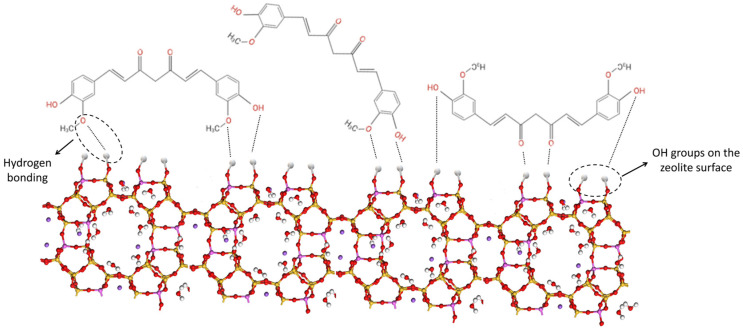
Schematic representation of hydrogen bonds between the OH groups from zeolite surface and the curcumin molecule. Original reproduction.

**Figure 8 pharmaceutics-15-01352-f008:**
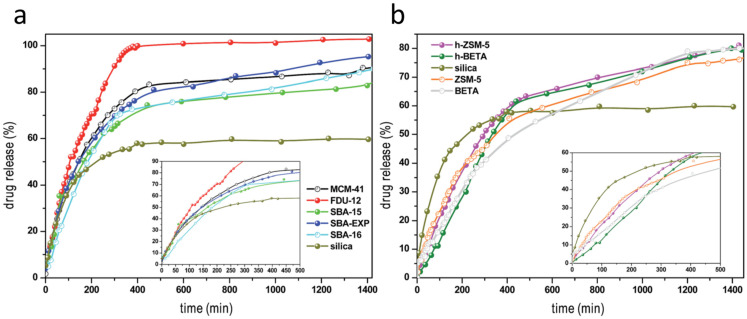
Drug release profile for methylprednisolone hemisuccinate loaded on (**a**) silica and mesoporous materials (MCM-41, FDU-12, SBA-15, SBA-EXP, SBA-15); (**b**) silica, zeolite materials (ZSM-5, BETA), and hierarchical zeolite (h-ZSM-5 and h-BETA). Adapted from [27], reproduced with published by ROYAL SOC CHEMISTRY, 2013.

**Figure 9 pharmaceutics-15-01352-f009:**
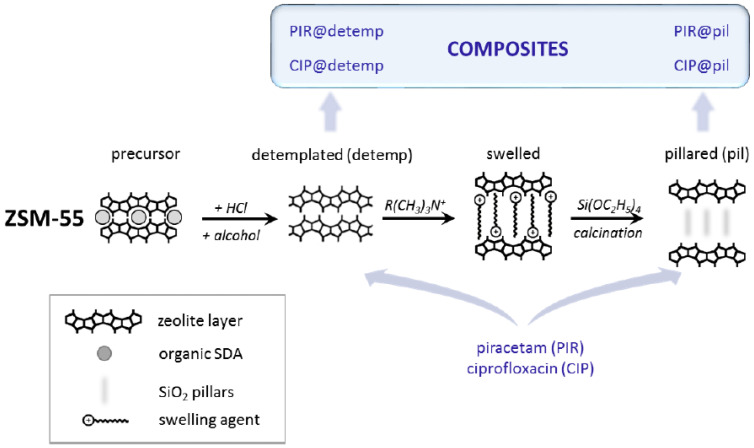
Schematic representation of ZSM-55 detemplation and pillarization prior to be used as piracetam and ciprofloxacin drug carrier. Reproduced with permission from reference [145], under creative commons license, published by MDPI, 2020.

**Table 1 pharmaceutics-15-01352-t001:** Pharmaceutical uses of unmodified, natural, and synthetic zeolites.

Zeolite Type	Pharmaceutical Uses	Treatment	Ref
Clinoptilolite	Active substance	Anti-diarrhoeic (Enterex^®^)	[44]
Clinoptilolite	Active substance	Antacid (Neutacid^®^)	[35]
Clinoptilolite	Active substance	Gastroesophageal reflux disease (Absorbatox^TM^)	[63]
Clinoptilolite	Active substance	Antiviral	[45]
Clinoptilolite	Active substance	Anticancer	[51,52]
Clinoptilolite	Active substance	Osteoporosis	[59]
Clinoptilolite	Carrier of zinc and erythromycin	Anti-acne	[71,72]
Clinoptilolite	Carrier for Acacia Catechu extract	gastrointestinal tract treatment	[87]
Clinoptilolite/ Mordenite	Carriers of copper ions	Antimicrobial activity	[93]
Faujasite	Danazol	Endometriosis, fibrocystic breast disease	[86]
Zn-clinoptilolite/ graphene oxide	Doxorubicin	Anticancer	[94]
Zeolite Y	Pyrantel pamoate and fenbendazole	Anthelmintic	[89]
Zeolite Beta	Salbutamol, theophylline	pulmonary disease	[95]
Zeolite Beta	Mitoxantrone	Anticancer	[96]
Zeolite Beta	Nifedipine	Hypertension and angina	[84]
Zeolite X and Y	Diclofenac sodium, piroxicam	Anti-inflammatory treatment	[6]
Zeolite X and zeolite A co-crystallization	Ketoprofen carrier	Inflammatory gastrointestinal tract treatment; reduce gastric side effects	[88]
Zeolite Beta, ZSM-5 and NaX	Indomethacin	Anti-inflammatory treatment of musculoskeletal disorders	[85]
Faujasite, zeolite A	α-Cyano-4-hydroxycinnamic acid	Anticancer	[97]
Zeolite 5A	Curcumin	Anticancer	[98]
ZSM-5	Curcumin	Anticancer	[99]
Zeolite Beta, NaX-FAU	5-Fluorouracil	Anticancer	[28]
ZSM-5, zeolite A, NaX	5-Fluorouracil	Anticancer	[42]
NaY-Faujasite, Lynde Type	5-Fluorouracil	Anticancer	[100]
Faujasite	O_2_ and CO_2_	Anticancer	[47]
MCM-41, SBA-15, FDU-12, SBA-16, h-ZSM-5, h-BETA	Methylprednisolone hemisuccinate	Rhinosinusitis	[27]
Zeolite X	Silver	Antimicrobial activity	[101]
Zeolite Beta, MFI and NaX	Probucol	Antilipidemic	[21]
Zeolite Beta, zeolite Rho, Paulingite	Silver	Antimicrobial, antitumor activity	[102]
Calcium-rich zeolite A and X	Risedronate	Osteoporosis treatment	[91]
NaX, NaY and HY	Clofazimine	Leprosy treatment	[90]
MFI	Lysozyme	Antimicrobial prophylaxis for implantable devices	[103]
MFI borosilicate	Doxorubicin	Anticancer	[104]

**Table 2 pharmaceutics-15-01352-t002:** Polymeric composites.

Zeolite Type	Polymer	Active Substance	Drug-Loading Method	Ref
Clinoptilolite	Chitosan	Diclofenac sodium, indomethacin	Solvent evaporation	[111]
Clinoptilolite	Chitosan, Polyvinil alcohol	Thymol	Coprecipitation	[76]
Clinoptilolite	poly(ε-caprolactone)-poly(ethyleneglycol)-poly(ε-caprolactone)	Bone tissue regeneration scaffold	-	[115]
Linde type L	poly-L-lysine	5-fluorouracil	Mixing	[110]
Zeolite Y and ZSM-5	Polyethylene glycol	Curcumin	Mixing	[112]
ZSM-5/KIT-6 and ZSM-5/SBA-15	Chitosan and carrageenan	verapamil	Impregnation	[113]
Zeolite-A	β-cyclodextrin	levofloxacin	Mixing	[114]
Zeolite Y and ZSM-5	Chitosan, poly lactic co-glycolic acid	Paclitaxel	Mixing	[116]
Natural zeolite type B	Poly(ethylene glycol) diacrylate	5-Fluorouracil	Immersion and ultrasonicated	[111]
Natural zeolite type B	Poly(ethylene glycol) diacrylate	Rose Canine extract	Stirring and sonicated	[117]
NaY	Chitosan	AgSD	Impregnation	[74]
NaX	PLA/Chitosan	Doxorubicin	Stirring	[118]
HZSM-5	Gelatin/agarose/zeolite	Pomegranate peel extract	Mixing	[79]
Zeolite A	Chitosan	Biomaterial scaffold for tissue engineering	-	[119]

**Table 3 pharmaceutics-15-01352-t003:** Roles of zeolites in tissue engineering.

Zeolite Type	Scope	Scaffold Production Technique	Zeolite Performance	Ref
not specified	Bone regeneration	Electrospinning	Mechanical properties were improved	[120]
clinoptilolite	Bone regeneration	Solvent-free powder compression/particulate leaching	Improved the mechanical properties	[115]
clinoptilolite	Bone regeneration	Particulate leaching/compression molding	Improve bone regeneration and promote repair	[121]
not specified	Bone regeneration	Electrospinning	Key role in osteoblastic physiology	[122]
ZSM-5	Bone regeneration	Freeze-drying	Superior mechanical, radiographic and histological properties	[123]
Zeolite-A	Bone Tissue	Freeze-drying	Better cell attachment and survival	[119]
VPI-7	Bone Tissue	extrusion 3D printer	Mechanical properties and antibacterial activity	[124]
Faujasite Y	In situ drug delivery and bone regeneration	Particulate leaching-freeze drying approach	Improved the scaffold mechanical characteristics, providing a pH-sensitive and a drug sustained release	[125]

## Data Availability

No new data were created.

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
