# Peer review of "Zeolites as Ingredients of Medicinal Products"

_pharmaceutics, 2023, doi:10.3390/pharmaceutics15051352_

Round 1

Reviewer 1 Report

The review presented by the authors are interesting. However, it must be improved before publication:

1. The authors must improve the english grammar and style, many typos are found in the manuscript.

2. The authors must include in the review different SEM images for the zeolites and correlate the morphology with the main drug delivery properties (for example, delivery time, charging efficiency etc.).

3.  It is not clear how the zeolites or their composites have been used in the literature for drug delivery. The authors must summarize in a table the times of delivery according to the pH values and explain the mechanisms.

4. The porosity plays an important role for the drug delivery of zeolites. The authors must explain how the different types of pores (meso-,micropore etc) affect the drug delivery properties of the zeolite. For completeness, the authors must cite the following reference, which is a review about the role of pores for the drug delivery:

a)Materials Science and Engineering: C Volume 96, March 2019, Pages 915-940 https://doi.org/10.1016/j.msec.2018.11.067

5. The authors must refer to other similar materials to explain why the zeolites is prefered for drug delivery. Please, refer to the following article as an example for this purpose and add it as background in the manuscript:

Applied Surface Science Advances Volume 13, February 2023, 100378

https://doi.org/10.1016/j.apsadv.2023.100378

6. Authors must summarize in a table the advantages and disadvantages of the zeolites as part of medical products (compare with previous publications)

Author Response

The review presented by the authors are interesting. However, it must be improved before publication:

  1. The authors must improve the English grammar and style, many typos are found in the manuscript.

Thank you for the suggestion. The paper was submitted to mdpi English improvement. (We are waiting this corrections).

  1. The authors must include in the review different SEM images for the zeolites and correlate the morphology with the main drug delivery properties (for example, delivery time, charging efficiency etc.).

Thank you for the suggestion. SEM images were added and related to the zeolitic crystal morphology before and after drug adsorption.

“More recently, faujasite zeolite, on X and Y forms, was studied for the first time, as carrier for 6-mercaptopurine (MERC), a chemotherapeutic drug used to treat various diseases such as blood cancer, inflammatory bowel disease or Crohn’s disease [144]. The current methods of administering this drug present poor bioavailability, short half-life in plasma and provokes side effects, such as bone marrow suppression and hepatotoxicity. To enable the adsorption interaction between the zeolite and the drug, faujasite was ion exchange with Zn2+ ions, been possible the formation of complexes with sulphur and nitrogen atoms from MERC (Figure 6-a). The results proved the ad-sorption of the drug without the evidence of drug precipitation and zeolite aggregation Figure 6-b, and 78% release of the drug following a controlled manner. Important aspect in the use of zeolites for this purpose is the non-destruction and aggregation of the particles and the non-precipitation of drugs. As seen in Figure 6-b, significant changes in the morphology of the zeolite crystal are not observed after the adsorption step, a behavior that collaborate to prove the effectiveness of these zeolites as carriers.

Figure 6. a) Influence of ion exchange on drug adsorption of MERC on faujasite zeolite, b) SEM images for zinc zeolite X and Y before (Zn-X and Zn-Y) and after sorption of MERC (Zn-X-MERC and Zn-Y-MERC). Modified with permission from [144] under creative commons license.”

  1. It is not clear how the zeolites or their composites have been used in the literature for drug delivery. The authors must summarize in a table the times of delivery according to the pH values and explain the mechanisms.

Thank you for the suggestion. Carrying out this bibliographical study, the great versatility of the use of zeolites as components in the formulation of medicinal products became quite evident. Among the factors that have been observed to affect this versatility are the large number of possible zeolites that can be applied and, for each case, the possibility of modifying parameters, such as acidity, porosity, functionalization with other elements, ion exchange, etc., to meet the specific needs of the desired pharmaceutical applicability. For this reason, it is necessary to evaluate case by case, considering both the drug and/or purpose, the type of zeolite and which parameters can be improved for a more viable performance. Therefore, it was preferable to approach and explain case by case reported in this review than to try to summarize in table form (a task that might not be reasonable given the specificities of the cases reported here). A new sentence on this aspect has been added to the conclusion.

“The development of new materials that can act efficiently in the improvement of drug formulations and treatments is a big challenge since it is a multidisciplinary area that covers several fields of knowledge. When approaching the possibilities of zeolite applications in this scenario, its versatility is evident, being able to act from active elements to as a function of carrying drugs in a specific location. It is also evident that many parameters must be considered when it comes to the use of zeolites for drug formulations and treatments, such as porosity, their composition in relation to possible isomorphic substitutions and thus the existence of exchangeable cations, among others. Moreover, the large number of possible zeolites that can be applied between natural and synthetic and, for each case, the possibility of functionalization to meet the specific needs of the desired pharmaceutical applicability, makes it necessary to study case by case, considering both the drug and/or purpose, the type of zeolite and which parameters can be improved for a more viable performance.”

  1. The porosity plays an important role for the drug delivery of zeolites. The authors must explain how the different types of pores (meso-,micropore etc) affect the drug delivery properties of the zeolite. For completeness, the authors must cite the following reference, which is a review about the role of pores for the drug delivery:

a)Materials Science and Engineering: C Volume 96, March 2019, Pages 915-940 https://doi.org/10.1016/j.msec.2018.11.067

Thank you for the suggestion. The porosity aspect was better explained and the suggested article cited

New sentence: “Pore size and geometry. According to the International Union of Pure and Applied Chemistry (IUPAC), pores are subdivided according to their internal diameter into micropores (> 2 nm), mesopores (between 2 and 50 nm) and macropores (< 50 nm) [23], in which zeolites generally present pores between the range of micro to mesopores. Depending on the drug molecular size, this pore opening and the geometry of the channels must be sufficient to allow access and diffusion within the zeolite structure in order to favor the encapsulation and release of different actives. In addition to porosity, surface area is an important parameter when it comes to adsorption and release of drugs. According to Esquivel-Castro et al., the increase in surface area is proportional to the amount of drug stored in core/shell nanoparticles [24]. These properties make possible to control the release and/or to protect certain molecules, such as nucleic acids, corticosteroids or anticancer drugs [25–28]. Nonetheless, these properties on its own usually fails to predict the drug-zeolite interaction, thus being necessary to consider the flexibility of the drug molecules and the zeolite framework in different conditions (such as high temperatures). In particular, Wernert and coauthors [29] observed that, during hydrothermal treatment, the pore volume of faujasite decreased in a directly proportional manner to the temperature, which subsequently influenced the entrance of the drug in the micropore. The use of structure-directing agents is another versatile mechanism for zeolite synthesis that uses a template molecule, organic or inorganic, as a model during the crystallization process, enabling the development of adequate porosity for the synthesized zeolite, creating a regular chain of pores and channels. After the crystallization, the template is removed. For instance, the addition of para-cresol during zeolite faujasite-Y syn-thesis resulted in a zeolite having an optimal pore size to retain this particular mol-ecule (and smaller ones) [30]. In addition, several methodologies can be applied be-fore or after the synthesis of the zeolite in order to create a secondary porosity, obtaining as a result hierarchical pores, expanding even more the versatility of the use of zeolites [31].”

  1. The authors must refer to other similar materials to explain why the zeolites is prefered for drug delivery. Please, refer to the following article as an example for this purpose and add it as background in the manuscript:

Applied Surface Science Advances Volume 13, February 2023, 100378

https://doi.org/10.1016/j.apsadv.2023.100378

Thank you for the suggestion. We understand that zeolites are another option for the formulation of medicinal products. We understand that several other materials are important and have revolutionized the medicinal product industry, such as polymers and clays, materials that were used as an example at various times in this manuscript. Making a comparison with other types of materials would be very relevant for this bibliographical survey, however we would need more than 10 days for this (time provided for corrections in the manuscript).

  1. Authors must summarize in a table the advantages and disadvantages of the zeolites as part of medical products (compare with previous publications)

Thank you for the suggestion. Summarize in a table all the advantages and disadvantages of the zeolites as part of medical products could be an easy way to correlate some of the informations on this review, but do that will require more de 10 days (established deadline for corrections) to finish.  In addition, the advantages and disadvantages of using zeolites as medicinal products will depend on each case, depending on the type of application, the type of drug, the physicochemical properties of these drugs, etc. for this reason, the applicability, advantages, results and behaviors were addressed throughout the manuscript, summarizing them when possible. We hope that the way in which this manuscript was written, taking into account the corrections suggested by the reviewers, is comprehensive and accessible to the scientific community.

Reviewer 2 Report

Recommendation: Minor Revision

Comments:

1.      The abstract should be elaborated to catch the scope of the review.

2.      Synthesis of zeolites should in included.

3.      Before showing the biomedical applications of zeolites, their properties should be explained. Also, the cytotoxicity of zeolites should be explained with the help of literature.

4.      The methods of preparing Zeolite/polymer composite should be explained with their advantages and limitations.

5.      The abstract should be elaborated to catch the scope of the review.

6.      The following paper may be helpful to this review article.

-      https://doi.org/10.3390/pharmaceutics11070305

Author Response

  1. The abstract should be elaborated to catch the scope of the review.

Thank you for the suggestion. The abstract was elaborated.

“Development of new medicinal products for particular therapeutic treatment or for better manipulations with better quality and less side effects are possible, among several factors, as a result of advanced inorganic and organic materials application, in which among them the zeolites, due to their properties and versatility, have been gaining attention. This paper is an overview of the development in the use of zeolite materials and their composites and modifications as medicinal products for several purposes such as active agents, as carriers, for topical treatments, oral formulations, as anticancer, in the composition of theragnostic systems, for vaccines, parenteral dosage forms, on tissue engineering, etc. The objective of this review is to explore the main properties of zeolites, associate them with their drug interaction, mainly addressing the advances and studies related to the use of zeolites for different types of treatments due to their zeolite characteristics, as molecule storage capacity, physical and chemical stability, cation exchange capacity and possibility of functionalization. The use of computational tools to predict the drug-zeolite interaction is also explored. As conclusion was possible to realize the possibilities and versatility of zeolite applications in this scenario, being able to act in several aspects of medicinal products.

  1. Synthesis of zeolites should in included.

Thank you for the suggestion. We agree that the synthesis of zeolites is an important topic, however exploring this topic in more detail is not the purpose of this review considering that there are several synthesis methodologies for zeolites. However, a new topic was added to the manuscript briefly addressing zeolite synthesis.

News sentences: “The synthetic zeolites can be advantageous due to the possibility to obtain zeolites with predetermined properties depending on the synthesis process and with less impurities. The synthesis process consists on prepare a mixture containing all the specific components correlated to each zeolite structure and submit this initial precursor mixture to hydrothermal treatment inside an autoclave at specific temperature to enable the nucleation and growth of zeolite crystal [33].

“A major part of the currently existing zeolites is of synthetic origin [51]. Zeolite synthesis consists in general on a hydrothermal crystallization process in alkaline me-dium. The preparation of initial precursor mixture containing the aluminosilicate sources enables the first polymerization and depolymerization processes, the hydro-thermal step leads the nucleation and crystallization process where the source under action of mineralizing agents and the structure-directing agent (SDAs) transforms into the zeolite crystal, as shown schematically in Figure 3 [33]. The optimization of differ-ent parameters such as temperature conditions, agitation, Si/Al ratio, among others, allow the synthesis of different zeolite structures [52,53].

Figure 3. Simplified scheme for the synthesis of a zeolite. Reproduced with permission from ref-erence [33]. Copyright© 2013 Elsevier Inc.”

  1. Before showing the biomedical applications of zeolites, their properties should be explained. Also, the cytotoxicity of zeolites should be explained with the help of literature.

Thank you for the suggestion. We agree that explore the zeolite proprieties is very important for their application on pharmaceutical formulations, proprieties that we cited during all the review, although there is several types of zeolites with different specific proprieties and explore more in detail this aspect is not the focus of this review, reason why the properties of zeolites were explored more generally and throughout the entire manuscript. The zeolite cytotoxicity was explained according to literature.

The cytotoxicity is one important parameter to be considered to application of new materials as ingredients for medicinal products. According to Bacakova et al. [34] some zeolites present an undesirable cytotoxicity, such as the case of erionite, but in some cases this cy-totoxicity can be advantageous for the treatment of tumors, that is, as well as several others parameters, a possible cytotoxicity of a zeolite must be studied case by case [34]. Studies report low toxicity for many types of zeolites, such as the adsorption, release and cytotoxicity of zeolite Y nanoparticles for the cisplatin, an anticancer medicine, were investigated and observed that the nano-zeolite did not show toxic effects on MG63 cells (supplemented with fetal bovine serum and streptomycin-penicillin) and exhibited a good biocompatibility, the composite ZC-NPs made by drug encapsulation showed a high cytotoxicity and an important reduction in cell viability [35]. The cytotoxicity activity of nanozeolite Y and A using alveolar epithelial cells (A549), hu-man endothelial cells (EA.hy926) and differentiated macrophages (THP-1) cell lines by mitochondrial activity (MTT) and cell membrane integrity (LDH leakage assay) were reported and observed no significant cytotoxic after 24h of exposure [36]. No significant toxicity were also observed for zeolites LTL and EMT [37], ZSM-5, zeolite A and Faujasite NaX [38].”     

  1. The methods of preparing Zeolite/polymer composite should be explained with their advantages and limitations.

Thank you for the suggestion. The methods of preparing Zeolite/polymer composite were explained.

“            The most used natural zeolite, clinoptilolite, has been combined with natural polymers to improve the physical properties of the latter. In particular, 3D polymer composites are susceptible to perform drug burst release due to their poor mechanical strength and high swelling ratio. A three-dimensional biocomposite based on chitosan was combined with clinoptilolite and loaded with indomethacin and diclofenac sodium, the biocomposite was prepared by the green method of cryogelation, on which ice crystals are acting as template preventing the formation of toxic byproducts or the use of severe chemical treatments for template extraction, for drug loading the solvent evaporation technique was performed [115]. As expected by the nature of the polymer, the drug release was pH-dependent, either with or without clinoptilolite. Nonetheless, the cumulative drug release was slower as higher the amount of clinoptilolite present in the composite (for both drugs).

Several publications report the formulation of nanocomposites based on zeolites and polymers for application as anticancer MDDS (see section 3.6.1), such as the syn-thesis and characterization of cross-linked poly(ethylene glycol) diacrylate nanogels containing natural zeolites, loaded with 5-fluorouracil (5FU), an efficient chemotherapeutic drug for several species of cancers such as breast, rectal and stomach cancers[116]. In this study the zeolite–nanogels composite, were obtained by an inverse mini-emulsion polymerization technique, a system that allow obtaining a stable mixture of water-soluble polymer micelles, composed of droplets of a polymer aqueous solution suspended by a mixture of surfactants and co-surfactants in a continuous organic medium [116].

Another example is the functionalization of Linde type L with poly-L-lysine to improve the cellular uptake of the composite and, therefore, improve the treatment efficacy, in this case the zeolite surface was modified via attachment with NH2 groups, COOH groups and after coated with the polymer [117]. Polyethylene glycol was used as a surface modifying agent of zeolites Y and ZSM-5 by mixing and the nanocomposites obtained studied as delivery system for curcumin [118] and, finally, the coating of zeolite nanoparticles with a three-layer polymeric shell of chitosan-k-carrageenan-chitosan as modified delivery system for verapamil [119].”

  1. The abstract should be elaborated to catch the scope of the review.

Thank you for the suggestion. The abstract was elaborated.

  1. The following paper may be helpful to this review article.

-      https://doi.org/10.3390/pharmaceutics11070305

Thank you for the suggestion.

Reviewer 3 Report

A good and broad review of the use of zeolites in drug delivery. A few minor points to be addressed prior to publication. 

Line 85: Zeolites contain Ions not atoms; an important distinction. 

Line 88: ammonia is not a positive molecule. The ammonium ion is positive and can be included as an ionic species.

Figure 1 is not referenced in the caption.

Line 124: replace schematised with 'shown schematically' 

Line 125: Low Si/Al ratio rather than small

Lines 136-139: The point / argument in this paragraph is not referenced.

Line173: The link between CEC and Al content should be emphasised here; it is mentioned, more in passing further down, but it is relevant at this point.

Lines 189 and 190 (and paragraph). Montmorillonite and palygorskite are smectite clays not zeolites. The paragraph does not mention zeolites hence this paragraph is somewhat out of context. It should perhaps be removed.

Is Figure 2 original or a reproduction? 

Line240: the Ammonia ion (please correct the notation to NH4+- the '4' should be subscript)  is not a gas.

Line 274: A high surface area clinoptilolite

Line330: tested not testes. 

Lines 332-335: Feel a little like a throwaway statement implying there's other information to substantiate the final statement but it is not provided in support of the statement. Expand or delete this paragraph.

Line 360: route not rout

Line 517 guarantee is incorrectly spelled 

Line 529: summarises (plural)

Figure 3 is not referenced 

Line 695: Remove 'The' at the beginning of the sentence. 

Line 805 route not rout

Line 809: Zeolite Beta rather than Beta Zeolite  

Figure 6 is not referenced 

Line 972: Zeolite Beta rather than Beta Zeolite

Author Response

A good and broad review of the use of zeolites in drug delivery. A few minor points to be addressed prior to publication.

  1. Line 85: Zeolites contain Ions not atoms; an important distinction.

Thank you. The sentence was modified.

New sentence: “The occurrence of silicon isomorphic substitutions by other elements such as aluminum creates a charge imbalance in the structure which is compensated by ex-changeable alkali and alkaline earth metal cations

  1. Line 88: ammonia is not a positive molecule. The ammonium ion is positive and can be included as an ionic species.

Thank you. The sentence was modified.

New sentence: “…positive molecules such as ammonium ion can also be associated with zeolites…”

  1. Figure 1 is not referenced in the caption.

Thank you. The figure one was created by the authors. A new sentence was included in the figure caption.

“Schematic illustration of the structure and components of a zeolite. Original reproduction.”

  1. Line 124: replace schematised with 'shown schematically'

Thank you. The sentence was modified.

New sentence: “…zeolite-based MDDS, as shown schematically in Figure 2.

  1. Line 125: Low Si/Al ratio rather than small

Thank you. The sentence was modified.

New sentence: “Low Si/Al ratios have demonstrated…

  1. Lines 136-139: The point / argument in this paragraph is not referenced.

Thank you for the correction. The references were included.

  1. Line173: The link between CEC and Al content should be emphasized here; it is mentioned, more in passing further down, but it is relevant at this point.

Thank you for the suggestion. The point was emphasized

New sentence: “Cation exchange capacity (CEC) and cations affinity. The CEC allow zeolites to es-tablish electrostatic interactions with cationic molecules by exchanging their natu-rally associated cations (such as Ca2+, Na+, Mg2+…) and positively charged molecules (ammonia and nitrate ions) with new cations or other positively charged molecules located in their surroundings [30]. Ion exchange is one of the most important prop-erty of zeolites, the isomorphic replacement of Si by elements with different charge, Al for example, become zeolite charged, which is counterbalanced by ions, named counterions, when the zeolite is immersed in a phase containing others ions in higher concentration the diffusion process is started, the initial ions diffuse out from the framework and the news ions diffuse into the solid structure [31]. This property makes zeolites to be used for the removal of toxins from wastewater and can be also explore on medicinal application. In fact, this property has been associated with antacid activity of clinoptilolite [32], which is able to exchange cations with the H+ of the gastric fluids. The affinity by certain types of cations should also be considered since it will determine the mobility of chemical species [33]. Moreover, different cat-ions could alter the adsorption/desorption capacity of organic molecules. As example, two potassium cations would be replacing one calcium, which could explain the variability in the adsorption capacity of zeolites with different exchangeable cations [27].”

  1. Lines 189 and 190 (and paragraph). Montmorillonite and palygorskite are smectite clays not zeolites. The paragraph does not mention zeolites hence this paragraph is somewhat out of context. It should perhaps be removed.

Thank you. This sentence was included only as an example of a natural inorganic material used industrially in the pharmaceutical industry, however, in order not to confuse the information, the suggestion was accepted and the sentence removed. 

New sentence: “The use of naturally occurring zeolites implies the presence of different minerals and impurities, as well as possible contaminants that must be eliminated (or, at least re-duced) before their use in the pharmaceutical field. In 2016, Cerri and co-workers published an article on the characterization and purification of clinoptilolite in order to accomplish with the requisites of the Japanese and European Pharmacopoeias [14].”

  1. Is Figure 2 original or a reproduction?

Yes. Figure 2 is an original reproduction. A new sentence was included in the figure caption.

Overview and classification of the main factors determining the selection of zeolites as drug delivery systems. Original reproduction.”

  1. Line240: the Ammonia ion (please correct the notation to NH4+- the '4' should be subscript) is not a gas.

Thank you. The sentence was modified.

New sentence: “Another useful therapeutic activity of zeolites in general, is their ability to adsorb molecules with dimensions capable of accessing its pores and cavities. In particular, clinoptilolite has been part of Panaceo® sport, a formulation intended to adsorb NH4+, CO2 and H2S…”

  1. Line 274: A high surface area clinoptilolite

Thank you. The sentence was modified.

New sentence: “A high surface area clinoptilolite (Panaceo Micro Activation or PMA-zeolite)…”

  1. Line330: tested not testes.

Thank you. The sentence was modified.

New sentence: “…silicalite-1 zeolite were tested as against C. auris…”

  1. Lines 332-335: Feel a little like a throwaway statement implying there's other information to substantiate the final statement but it is not provided in support of the statement. Expand or delete this paragraph.

Thank you for the suggestion. The paragraph was deleted.

  1. Line 360: route not rout

Thank you. The sentence was modified.

New sentence: “Synthetic ZSM-5 zeolite was synthetized through hydrothermal route and pro-posed…”

  1. Line 517 guarantee is incorrectly spelled

Thank you. The sentence was modified.

New sentence: “In order to guarantee a high amount of calcium…”

  1. Line 529: summarises (plural)

Thank you. The sentence was modified.

New sentence: “The table 1 summarises some of unmodified natural and synthetic zeolite…”

  1. Figure 3 is not referenced

The figure 2 is an original reproduction. A new sentence was included in the figure caption.

New sentence: “Zeolite surface modification with surfactant and interaction with drugs molecules. Original reproduction”.

  1. Line 695: Remove 'The' at the beginning of the sentence.

Thank you. The sentence was modified.

New sentence: Osteosarcoma is one bone tumor which causes damage to nearby tissues.”

  1. Line 805 route not rout

Thank you. The sentence was modified.

New sentence: “Following a ship-in-a bottle route…

  1. Line 809: Zeolite Beta rather than Beta Zeolite

Thank you. The sentence was modified.

New sentence: “Zeolite beta was evaluated as mitoxantrone carrier”

  1. Figure 6 is not referenced

The figure 6 is an original reproduction. A new sentence was included in the figure caption.

New sentence: “Schematic representation of hydrogen bonds between the OH groups from zeolite surface and the curcumin molecule. Original reproduction.”

  1. Line 972: Zeolite Beta rather than Beta Zeolite

Thank you. The sentence was modified.

New sentence: “The zeolite beta with Si/Al ratio = 37…

Reviewer 4 Report

This is a very readable popular science paper on zeolite application in medicine. The following issues need to be addressed,

1.         Zeolite is a broad concept, and there are more than 250 kinds of synthetic zeolite alone. In this manuscript, the application of natural or synthetic zeolites, such as, clinoptilolite and NaA, NaX, NaY, Beta, ZSM-5, is reviewed. Actually, which zeolite is used in what respect is still unclear, which performance of the zeolite is required in the medicinal products is also unclear.

2.         The application of natural or synthetic zeolites in oral administration formulations. How the zeolite is excreted because of the water insolubility and imdecomposition of the zeolite in the body?

3.         The functionalization/grafting on the zeolite surface is very difficult because of few OH groups resulted from zeolitic crystalline structure, esp. high-silica zeolite such as ZSM-5 (Si/Al ratio: 12-) and Beta.

4.         Line 268-269, “The zeolite was previously activated by a tribomechanical process, which produced nanoparticles with a surface of 50,000 m2 and increase the specific surface area.” Here there is no concept of quantity, and the specific surface value needs to be given.

Author Response

This is a very readable popular science paper on zeolite application in medicine. The following issues need to be addressed,

  1. Zeolite is a broad concept, and there are more than 250 kinds of synthetic zeolite alone. In this manuscript, the application of natural or synthetic zeolites, such as, clinoptilolite and NaA, NaX, NaY, Beta, ZSM-5, is reviewed. Actually, which zeolite is used in what respect is still unclear, which performance of the zeolite is required in the medicinal products is also unclear.

Thank you for the suggestion. The large number of possible zeolites that can be applied on medical products and the possibility of modifying parameters, such as acidity, porosity, functionalization with other elements, ion exchange, etc., to meet the specific needs of the desired pharmaceutical applicability becomes difficult to establish which zeolite, which parameters and performance are more relevant. A new sentence has been added to the conclusion addressing this aspect.

“The development of new materials that can act efficiently in the improvement of drug formulations and treatments is a big challenge since it is a multidisciplinary area that covers several fields of knowledge. When approaching the possibilities of zeolite applications in this scenario, its versatility is evident, being able to act from active elements to as a function of carrying drugs in a specific location. It is also evident that many parameters must be considered when it comes to the use of zeolites for drug formulations and treatments, such as porosity, their composition in relation to possible isomorphic substitutions and thus the existence of exchangeable cations, among others. Moreover, the large number of possible zeolites that can be applied between natural and synthetic and, for each case, the possibility of functionalization to meet the specific needs of the desired pharmaceutical applicability, makes it necessary to study case by case, considering both the drug and/or purpose, the type of zeolite and which parameters can be improved for a more viable performance.”

  1. The application of natural or synthetic zeolites in oral administration formulations. How the zeolite is excreted because of the water insolubility and indecomposition of the zeolite in the body?

Thank you for the suggestion. During this bibliographic survey, it was observed that many studies related to oral administration are carried out in the in vitro phase, and those that have already been published in vivo did not address this aspect. This topic is a very interesting topic and deserves a lot of attention however would necessary more than the time determined from the journal to do the corrections suggested for the reviewers.

  1. The functionalization/grafting on the zeolite surface is very difficult because of few OH groups resulted from zeolitic crystalline structure, esp. high-silica zeolite such as ZSM-5 (Si/Al ratio: 12-∞) and Beta.

Thank you for the suggestion. The existence of different types of zeolites with different Si/Al ratios, in addition to the possibility of ratio changes for some types of zeolites, is an aspect that must be taken into account when carrying out functionalization in zeolites. Addressing the possible functionalization of zeolites with a high silica applied to the formulation of medicinal products would be quite interesting, but we would need more than 10 days for this study, not meeting the requirements of the journal.

  1. Line 268-269, “The zeolite was previously activated by a tribomechanical process, which produced nanoparticles with a surface of 50,000 m2 and increase the specific surface area.” Here there is no concept of quantity, and the specific surface value needs to be given.

Thank you for the suggestion. the reference was reanalyzed and the sentence rewritten.

New sentence: “The zeolite was previously activated by centrifugal process on a tribomechanical micronization and activation device, which produced nanoparticles with total surface of 50,000 m2 and particle size of 200 nm. In this occasion, clinoptilolite acted as an immune-stimulator that helped scar tissue formation. Additionally, UV and antibacterial protection was also provided by clinoptilolite, which are very convenient performances of an ingredient during the wound healing process [53,54].

Round 2

Reviewer 1 Report

The article can be accepted for publication.